# A binary interaction map between turnip mosaic virus and *Arabidopsis thaliana* proteomes

Fernando Martínez[1,4], José L. Carrasco [1,4], Christina Toft[1], Julia Hillung[1], Silvia Giménez-Santamarina [2], Lynne Yenush [2], Guillermo Rodrigo [1,4✉] & Santiago F. Elena [1,3✉]

Viruses are obligate intracellular parasites that have co-evolved with their hosts to establish an intricate network of protein–protein interactions. Here, we followed a high-throughput yeast two-hybrid screening to identify 378 novel protein–protein interactions between turnip mosaic virus (TuMV) and its natural host *Arabidopsis thaliana*. We identified the RNA-dependent RNA polymerase NIb as the viral protein with the largest number of contacts, including key salicylic acid-dependent transcription regulators. We verified a subset of 25 interactions *in planta* by bimolecular fluorescence complementation assays. We then constructed and analyzed a network comprising 399 TuMV-*A. thaliana* interactions together with intravirus and intrahost connections. In particular, we found that the host proteins targeted by TuMV are enriched in different aspects of plant responses to infections, are more connected and have an increased capacity to spread information throughout the cell proteome, display higher expression levels, and have been subject to stronger purifying selection than expected by chance. The proviral or antiviral role of ten host proteins was validated by characterizing the infection dynamics in the corresponding mutant plants, supporting a proviral role for the transcriptional regulator TGA1. Comparison with similar studies with animal viruses, highlights shared fundamental features in their mode of action.

[1] Instituto de Biología Integrativa de Sistemas (I2SysBio), CSIC - Universitat de València, 46980 Paterna, Spain. [2] Instituto de Biología Molecular y Celular de Plantas (IBMCP), CSIC - Universitat Politècnica de València, 46022 València, Spain. [3] The Santa Fe Institute, NM87501 Santa Fe, NM, USA. [4]These authors contributed equally: Fernando Martínez, José L. Carrasco, Guillermo Rodrigo. ✉email: guillermo.rodrigo@csic.es; santiago.elena@csic.es

Viral infections represent an important threat in our global society, not only because some can greatly compromise human health[1], but because some jeopardize intensive agriculture and livestock production[2], both which are of special relevance in the current scenario of climate change and alimentary crisis[3]. A correct understanding of the different molecular mechanisms that contribute to the burden imposed by viral infections is instrumental for the successful development of treatments and control measures. In this regard, the identification of host factors that directly interact with viral components has long been one of the central goals of molecular virology[4]. However, with the advent of high-throughput omics techniques[5], the approach has shifted from the identification and thorough characterization of individual host factors to the simultaneous experimental determination and network-based computational analyses of hundreds of virus-host interactions[6,7]. Certainly, by defining how viruses perturb the host proteome in terms of function and structure, we can gain a holistic understanding of infection and its consequences for host physiology. However, the field of systems virology[6] still requires much work in order to grasp the yet unknown mode of action of many plant and animal viruses.

Viruses exhibit, through a limited number of proteins (especially reduced in the case of RNA viruses)[8], multiple contact points with the host proteome, i.e., virus-host protein–protein interactions (PPIs). Indeed, this is the combined result of, on the one hand, their necessity to manipulate diverse cellular pathways to create a favorable environment for their replication (either by sequestering resources for their own benefit or by interfering with the host immune responses) and, on the other hand, the ability of the host cell receptors to sense foreign elements and then respond accordingly. Considerable progress has been made over the last years to generate detailed, high-quality virus-host PPI maps in the case of human viruses[9–13]. Furthermore, integrative approaches have identified general and specific molecular mechanisms employed by different human viruses[14,15] and, together with additional omics data, have even been used to predict phenotypic outcomes of infection[16]. Certainly, all these developments take advantage of the continuous elaboration of an accurate large-scale map of human PPIs[17], which despite its undeniable incompleteness appears to be useful to recognize disease-associated modules[18].

According to previous network-based analyses, viral targets tend to be more connected with other host proteins than expected by chance[9,14]. This means that viral proteins interact with some hub proteins of the host interactome, which are seldom affected by randomly removing nodes[19], since biological networks are scale-free (i.e., there are a few elements strongly connected in the network, while most of them are weakly connected)[20]. Consequently, by preferentially affecting hub nodes, viruses perturb the host network at a global scale (and consequently host physiology) in a nonrandom manner, although significantly less than through pure centrality-directed attacks[21]. However, this property reflects an average tendency, as viral proteins also interact with a substantial set of non-central and even peripheral proteins. Previous studies have revealed that some viral proteins bridge different subnetworks within the host interactome that otherwise might appear as disconnected, with an increased ability to spread information[22]. In addition, we now realize that host proteins targeted by viral proteins, at least a subset of them, participate in signaling pathways that are linked to the symptoms of infections and are located in the neighborhood of disease-associated proteins, (e.g., genes with differential expression upon infection)[23]. Importantly, those with a marked function or connectivity in the host tend to be determinants of infection (either as required by the virus, i.e., proviral; or for the host to mount the antiviral immune response, i.e., antiviral)[11–13]. Furthermore, viral proteins showing the largest number of contacts are typically non-structural (i.e., do not form the virion) and have a relevant enzymatic activity[9,13]. From an evolutionary point of view, much controversy exists regarding whether viral targets display faster rates of adaptive evolution, which would indeed be indicative of a coevolutionary arms race between the virus and the host[24,25]. Quantitatively, one-third of the adaptive mutations in humans appear to be in response to viruses[25].

Because the study of plant viruses has not progressed at the same pace as human virology, there are still multiple unanswered questions regarding their mode of action, from both molecular and holistic perspectives[26]. While various studies have unveiled physical contacts between individual virus-host protein pairs that are relevant in terms of host infection[27–33], a systematic analysis of a virus-plant interactome is still lacking. Moreover, large-scale studies that have been carried out with cellular pathogen effectors in plants reveal a mode of action similar to the animal viral proteins, in network terms. These studies show a similar enrichment in central host proteins that interact with pathogen-effector proteins in both plants and animals[34,35]. Therefore, systems level analyses of plant virus-host PPI networks are required to advance the field of plant molecular virology and also to uncover general similarities and differences between animal and plant viruses that could contribute to our understanding of fundamental viral pathogenesis mechanisms.

In this work, we performed a systematic and stringent identification, using high throughput yeast two-hybrid (HT-Y2H) screening techniques[36,37], of the direct PPIs established between a plant virus and one of its natural hosts. We used the turnip mosaic virus strain YC5 (TuMV; species *Turnip mosaic virus*, genus *Potyvirus*, family *Potyviridae*) as a model system[26]. Potyviruses are the largest and most abundant family of plant RNA viruses in nature and are responsible for important crop losses[26]. As the host plant, we used *Arabidopsis thaliana* (L.) Heynh, the quintessential model in plant biology, with its repertoire of genetic tools, which is naturally infected by TuMV[38]. Here, we present our analyses of the identified virus-targeted host proteins in terms of their biological function, proteomic context, expression level, role as proviral or antiviral factors, and evolutionary constraints. Finally, we discuss our results in the light of data regarding phenotypic outcomes of infection in wild-type and mutant plants and by comparing them with the results reported for clinically-relevant human RNA viruses. Our results represent a valuable resource to move forward in the study of plant viruses from a systems biology perspective[39].

## Results

**Construction of a TuMV-*A. thaliana* PPI network.** We performed a HT-Y2H screening for each of the eleven proteins encoded in the TuMV genome against all *A. thaliana* proteins. For that, different clones expressing the virus proteins were constructed and a universal library containing normalized amounts of cDNAs from transcripts isolated from different plant tissues at different developmental stages was used. The screening was done through two mate-and-plate steps (first with a permissive medium to capture putative interactions, then with a more stringent medium to minimize false positives). Library plasmids were rescued from those clones identified as positives after the second round of screening and retransformed into yeast for validation of reporter gene activation, both in the presence and in the absence of the viral protein used in the screen (this was done for both nuclear and cytoplasmic virus proteins). Only those clones failing to activate reporter gene expression on their own

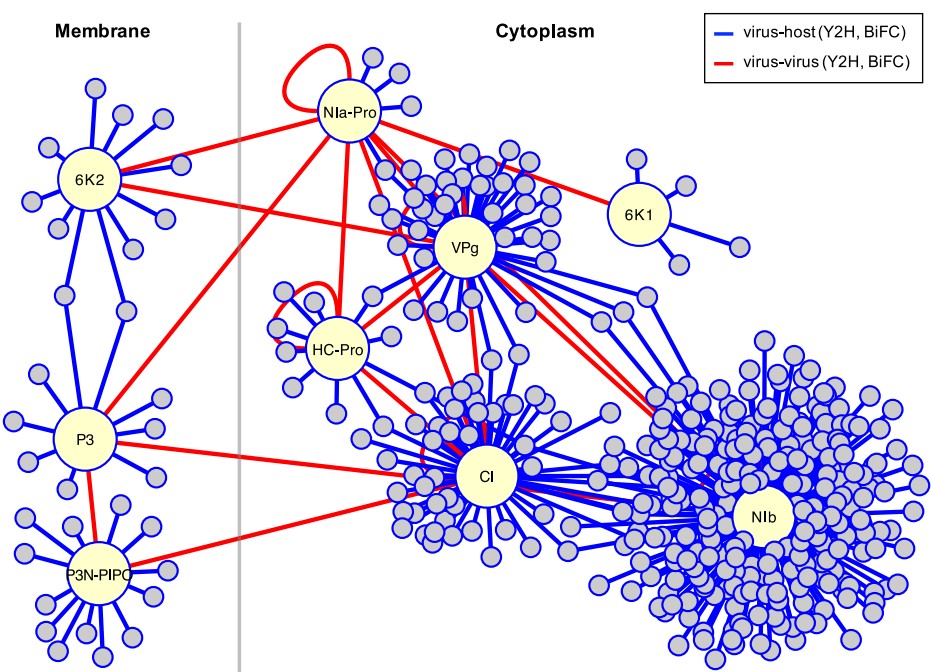

**Fig. 1 Virus-host protein-protein interactome constructed in this work by yeast two-hybrid screening.** This corresponds to the interaction of the plant virus TuMV and the host plant *A. thaliana* (blue edges). Virus proteins highlighted. The interactors of P3, P3N-PIPO, and 6K2 were retrieved with a library of *A. thaliana* proteins expressed in the membrane, whilst the interactors of HC-Pro, 6K1, CI, VPg, NIa-Pro, and NIb with a library of proteins expressed in the cytoplasm. Virus–virus protein–protein interactions (red edges) already reported in previous work complement this network view. The complete list of host proteins is provided in Supplementary Data 1 and Supplementary Data 2.

were selected for further analysis. As a result, we obtained 10 unique interactors for HC-Pro, 4 for 6K1, 54 for CI, 33 for VPg, 4 for NIa-Pro, and 245 for NIb. However, we did not obtain positive interactions in the case of P1 (no colonies at all) and CP (no colonies with the appropriate phenotype). Thus, P1 and CP were removed from the study. We then performed a second HT-Y2H screening capable of identifying interactors of membrane-associated proteins, i.e., with the split-ubiquitin (sUbq) system[40] using the same procedure to remove false positives (autoactivators). This was done for P3 (it has been previously shown to attach to the ER membrane system[26]), P3N-PIPO, and 6K2. This second screen resulted in the identification of 9 unique interactors for P3, 12 for P3N-PIPO, and 10 for 6K2. Collectively, our experiments identified 381 virus-host PPI, 378 of them described for the first time (Supplementary Data 1). Poty-viral NIb, the RNA-dependent RNA polymerase, is the central element in the viral replication complex (VRC) responsible for genome replication, showed the largest number of contacts with the host. NIb has already been described as the most promiscuous potyviral protein, as it actively recruits many pro-viral host proteins into the VRC[41].

To evaluate the sensitivity of our HT-Y2H assays, we compiled a positive reference set[42,43] of 58 potyvirus-plant binary PPIs described in the literature for different potyviruses and plant hosts[44]. Forty of them were detected during our screenings, thus resulting in 68.97% assay sensitivity. The 18 missed interactors, perhaps as a consequence of using a selection medium with increased stringency, were all described for the TuMV-*A. thaliana* pathosystem: 10 additional interactors for VPg, 6 for NIb, 1 for P3N-PIPO, and 1 for 6K2. Therefore, we decided to incorporate them to generate a final network with 399 virus-host PPI (Fig. 1 and an extended version including interactions among the 399 host proteins and their one-step neighbors in Supplementary Fig. 1).

The number of interactions into which a given protein can be involved has been related to its propensity to contain intrinsically disorder regions (IDPR)[45]. We have evaluated whether the number of interactions found for each TuMV protein depended on their IDPR. To do so, we first we evaluated IDPR using the fIDPnn server[46] and found that the number of IDPR varies among potyviral proteins, with CI and NIb showing the lowest propensity and 6K1 and VPg the highest (Supplementary Fig. 2). However, no correlation exists between the different indexes of IDPR computed by fIDPnn and the observed number of interactions, thus ruling out the possibility that our results can be solely explained by differences in the prevalence of IDPRs among TuMV proteins. Indeed, NIb and CI, the two proteins for which we found more interactors, show no propensity to contain IDPR. In sharp contrast, P1 and CP, the two proteins that we failed to identify interactors show strong evidence for IDPR.

To validate that some of the binary interactions found by the HT-Y2H screening indeed occur *in planta*, we performed bimolecular fluorescence complementation (BiFC) assays[47,48] in *Nicotiana benthamiana* Domin. For that, we selected a random subset of 25 virus targets. Twenty-four of them were novel, and only the interaction between VPg and the eukaryotic translation initiation factor eIF(iso)4E was already described for TuMV[27]. Notably, we observed specific reconstituted fluorescence in all tested pairs (Fig. 2). Among the confirmed ones, the SGS domain-containing protein encoded in locus *AT1G30070* was the most promiscuous one, interacting with HC-Pro, CI, and NIb. RUBREDOXIN-LIKE SUPERFAMILY PROTEIN was confirmed to interact with CI and NIb and DEHYDROASCORBATE REDUCTASE 3 (DHAR3) and METHYL-CPG-BINDING DOMAIN PROTEIN 5 (MBD5) both interacted with CI and VPg. All these interactions likely occur within the VRC, since these four viral proteins have been shown to be part of it[41]. The hypothetical protein encoded by locus *AT1G11125* also has been

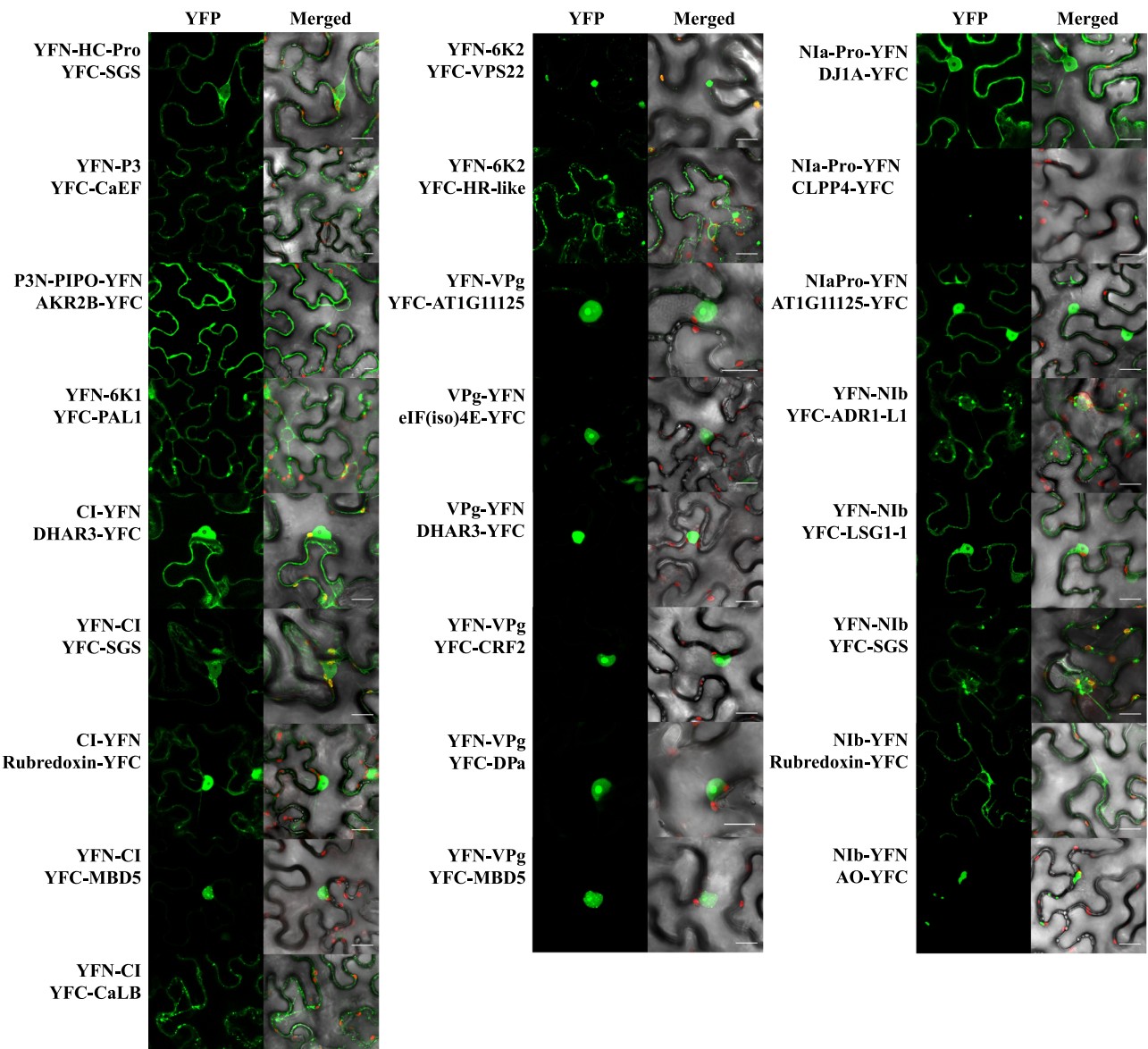

**Fig. 2 Validation of some virus-host protein–protein interactions by bimolecular fluorescence complementation *in planta* with a split YFP system.** The TuMV proteins were cloned with the YFP N-terminus and the host proteins with the YFP C-terminus. Confocal images of plant tissues (from *N. benthamiana*) to reveal the interaction by fluorescence (GFP filter), together with merged images (GFP and RFP filters) to localize the chloroplasts due to their red autofluorescence. Pictures of the heterologous plasmid controls showed no fluorescence signal. Scale bar, 20 μm.

confirmed to interact with VPg and NIa-Pro, suggesting that the interaction may actually occur with the NIa precursor. All other validated interactions involved NIb, indicating that also the many interactions of this protein validate at a high rate (Fig. 2).

**Functional analysis of the *A. thaliana* proteins targeted by TuMV.** As a starting point to study the effect of the perturbation introduced by the virus infection on host physiology, we assessed a potential enrichment of certain biological processes within the list of TuMV targets. For that, we took advantage of gene ontology (GO) resources[49]. We found that "response to stress" (and "response to virus" in particular), "post-transcriptional regulation of gene expression", "meristem development", or "photosynthesis" are among the 272 biological processes identified as significantly over-represented (Fig. 3a; Fisher exact tests for 2 × 2 contingency tables; Benjamini–Hochberg false discovery

rate (FDR) adjusted $P < 0.05$) (Notice these results are robust to removing the 18 literature-curated from the analyses, with the same functional categories popping up as significantly enriched). TuMV is a castrating virus that induces dwarfism, arrest of development of reproductive tissues and, strong chlorosis that may result in generalized necrosis and plant death. In consequence, the activation of stress responses along with genome-wide alterations in gene expression affecting developmental processes and photosynthesis illustrates the conflict between the host and the virus, which counteracts such defense and creates a favorable context for replication[50]. More specifically, we considered a relevant set of GO terms to be mapped against each virus protein (Fig. 3b). We observed that some TuMV proteins, despite having contacts with few host proteins (e.g., HC-Pro, P3N-PIPO, or 6K2), appear to perturb very different processes, from metabolism to regulation to defense. Certainly, this can be a consequence of targeting plant proteins that participate in

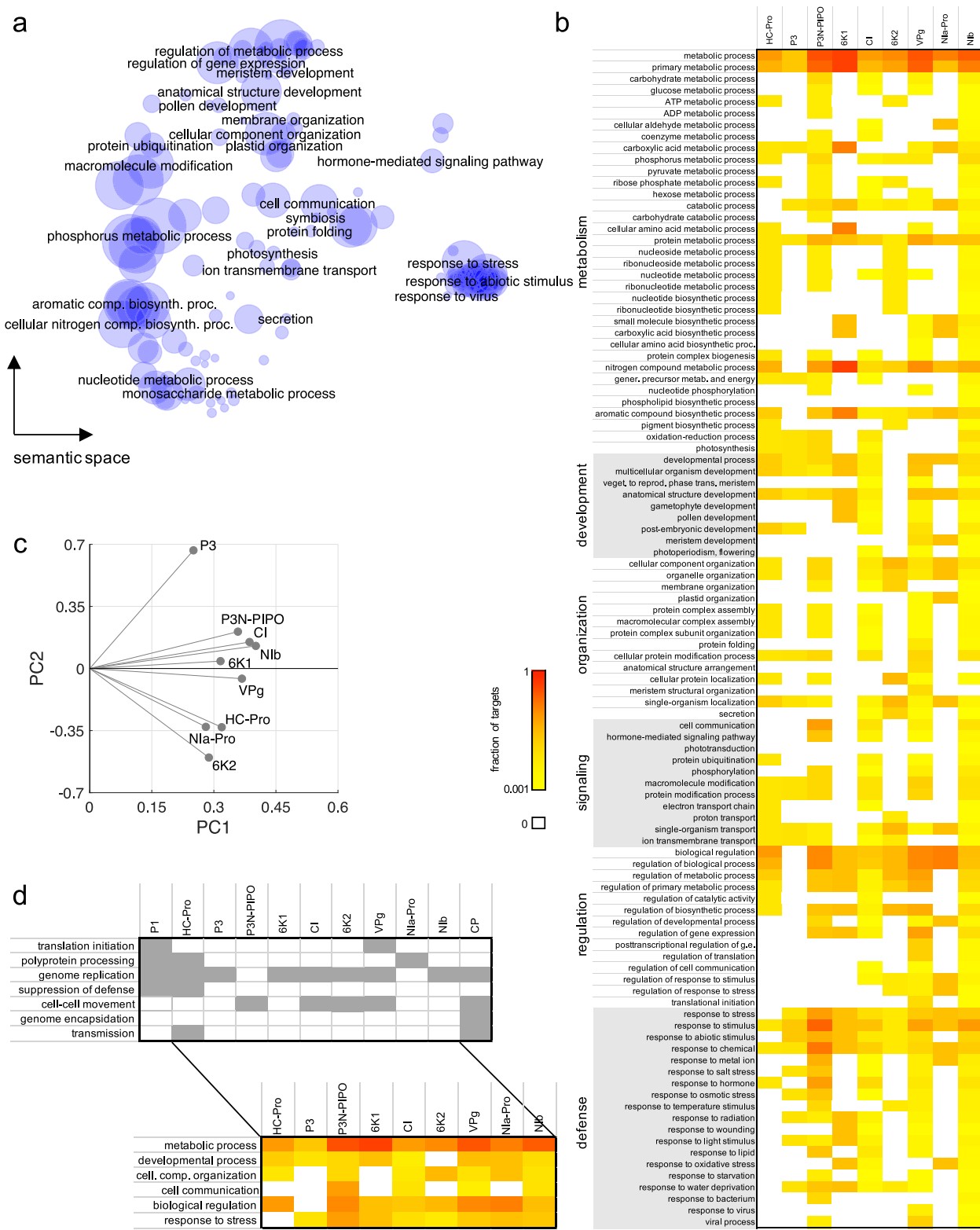

**Fig. 3 Functional analysis of the host proteins targeted by the plant virus TuMV. a** Representation in a semantic space of the functional categories (related to biological processes) enriched within the virus targets. Bubble size scales with the total number of proteins per category. Statistical significance assessed by Fisher exact tests, 2 × 2 tables, FDR adjusted *P* < 0.05. **b** Heat map for the main functions (related to metabolism, development, organization, signaling, regulation, and defense) targeted by each TuMV protein. Color scale indicates fraction relative to the total number of targets of that virus protein. **c** Principal component analysis of the data shown in panel **b**. The different virus proteins are distributed in this space according to the similarity of the functional overlap of their targets. **d** Associative mapping between the function of each virus protein (in the context of viral processes) and the main functions of each virus target (in the context of plant physiology).

multiple functions, such as PLASMA-MEMBRANE ASSO-CIATED CATION-BINDING PROTEIN 1 (PCaP1), targeted by P3N-PIPO[32,33], which reflects the highly intricate (mostly hormone-mediated) nature of the signaling and regulatory pathways in plants[51]. Of course, NIb, with the largest number of contacts, is the virus protein that appears to impacts more host functions. The heatmap also revealed functions targeted by (almost) all virus proteins, such as "response to stress", as well as functions with a more specific relationship, such as "translation regulation" (only targeted by VPg and NIb).

In addition, a principal component analysis with the data matrix shown in Fig. 3b organized the virus proteins in a two-dimensional space according to the similarity of their predicted impact on host physiology (Fig. 3c; with 74.2% of explained variance). Interestingly, we observed that virus proteins that physically interact (as indicated by the red lines in Fig. 1) are closer in this space, which may be indicative of a strategy evolved by the virus to more efficiently coordinate the action of its proteins during infection[52]. This is the case, for instance, for HC-Pro and NIa-Pro, VPg and NIb, or P3N-PIPO and CI. This analysis also highlights P3 targets as the most unique set of host proteins compared to the rest of the virus proteins. Finally, we compared the functions of the different virus proteins[26] with the functions of their host targets (Fig. 3d). For example, P3N-PIPO and CI are involved in cell-to-cell movement of viral particles. To achieve this goal, they affect host functions related with "organization of cellular compartments" and "cell communication". Similarly, 6K2 is required for constitution of the VRC and its attachment to ER membranes. Consequently, its interactors are mostly enriched in proteins whose functions are relevant for the "organization of cellular compartments". Indeed, we confirmed previous findings that interaction of 6K2 with SECRETION 22 (SEC22) SNARE is essential for the trafficking of replication vesicles via prevacuolar compartments[53]. In essence, the virus requires the exploitation and disruption of multiple biological processes in the host to complete its replication cycle, doing so by employing a set of multifunctional proteins.

**Topological contextualization of the *A. thaliana* proteins targeted by TuMV.** To evaluate the systems-level relevance of each identified virus target, we performed a network analysis. Firstly, we adopted a virus-centric approach to analyze how the virus proteome establishes interactions with the host proteome (Fig. 4a). We found a significant enrichment in host factors targeted at least by two different virus proteins (Fig. 4b; $z$ test, $P < 0.0001$), which suggests that such host factors can serve as edges between otherwise disconnected virus proteins[10]. For example (Supplementary Data 1), P3 and 6K2 both interact with the products of genes *AT2G20920* (the chaperone DUF3353) and *AT3G10840* (an α/β-hydrolase superfamily protein associated to chloroplast external membranes) or the SGS domain-containing protein, mentioned above, that may simultaneously form a complex with HC-Pro and NIb inside the VRC (Fig. 4a). While the length of the shortest paths that connect any two virus proteins is invariant whether or not the host proteome is considered (Fig. 4c), the number of paths increases appreciably when it is considered (Fig. 4d). Of special note is the emergent relationship that is established between the virus helicase CI and replicase NIb, which may reflect the necessity for coordination in the VRC[26,52].

Secondly, we adopted a host-centric approach to contextualize the virus targets into the AI-1$_{MAIN}$ network model[54] (Supplementary Data 3; Fig. 5a). We focused our analysis on two relevant topological properties: connectivity degree (i.e., the number of direct interactions that a given protein establishes with others) and average shortest path length (i.e., the average length of all shortest paths that connect a given protein with the rest). We found that the connectivity degree distribution for all host proteins has a scale coefficient greater than for the virus targets (as illustrated by the different slopes of the power laws that fit the data in Fig. 5b), which is indicative of an enhanced probability of the virus proteins to interact with host hub proteins. In particular, the scale coefficient dropped from 1.247 to 0.860, a reduction of 31.0% (comparison of slopes in an ANCOVA test, $F_{1,21} = 9.70$, $P = 0.0053$). A similar trend has been reported in the case of several human viruses[9–11,55]. In addition, by comparing the distributions of connectivity degrees and the average shortest path lengths for the virus targets and for different sets of random genes, we found that the distributions associated with the virus targets are significantly shifted towards higher connectivity degrees (Fig. 5c; Mann–Whitney $U$ test, $P = 0.0045$) and lower path lengths (Fig. 5d; Mann–Whitney $U$ test, $P = 0.0003$). For the virus targets, the mean of the degrees was 7.12 and the mean of the path lengths 4.18. These results are also observed upon removal of the 18 interactors gathered from the literature (Supplementary Fig. 3).

For illustrative purposes, we sketched the virus-host PPI by selecting three virus targets with high connectivity degree in the host interactome: the small ubiquitin-like modifier (SUMO) ligase SUMO CONJUGATING ENZYME 1 (SCE1), with 162 interactors (the target with the highest connectivity degree), the PHD finger OBERON 1 (OBE1), with 61 interactors, and the bZIP transcription factor TGACG SEQUENCE-SPECIFIC BINDING FACTOR 1 (TGA1), with 38 interactors (Fig. 5e). These proteins are targets of VPg and NIb. For completeness (as it interacts with both NIb and TGA1), the salicylic acid (SA)-dependent transcriptional regulator NONEXPRESSER OF PATHOGENESIS-RELATED GENES 1 (NPR1), which controls the expression of genes that exert a response against pathogens, is also shown (note that TGA1 shares interactors with NPR1 and OBE1 though they are not physically connected in the AI-1$_{MAIN}$ network). These interactions promote virus replication and within-host movement (i.e., proviral factors) and block host defenses (i.e., antiviral factors)[28,30,31,35]. A good example of such master regulator protein could be the CALCIUM-BINDING EF-HAND FAMILY PROTEIN (CaEF), a $Ca^{2+}$ sensor and partner of P3 (Supplementary Data 1), that participates in the orchestration of jasmonic- and SA-dependent signaling responses to bacterial and fungal infections as well as in responses to oxidative stress and is associated into the endomembrane system[56].

**Expression features of the *A. thaliana* proteins targeted by TuMV.** Next, we sought to study the expression levels of the virus targets, as well as the relationship between expression and connectivity within the *A. thaliana* interactome. For that, we collected the expression values of all *A. thaliana* genes in control conditions and also upon infection with TuMV from previous transcriptomic experiments[57]. By dividing the expression levels of healthy plants into three categories (low, medium, and high), we found virus targets in all of them, but with an apparent enrichment in the category of high expression (Fig. 6a). Note, for example, the ratio of highly vs. lowly expressed proteins in the case of P3N-PIPO (7 *vs.* 3, out of 13 host interactors), CI (28 *vs.* 6, out of 54 host interactors), or NIb (101 vs. 42, out of 251 host interactors). To further explore this issue, we compared the distributions of expression for the virus targets and for different sets of randomly selected genes, revealing that the distribution associated with the virus targets is significantly shifted towards higher expression levels (Fig. 6b; Mann–Whitney $U$ test, $P < 0.0001$). We repeated the comparison from expression data upon TuMV infection, finding a similar trend (Fig. 6c; Mann–Whitney $U$ test, $P < 0.0001$).

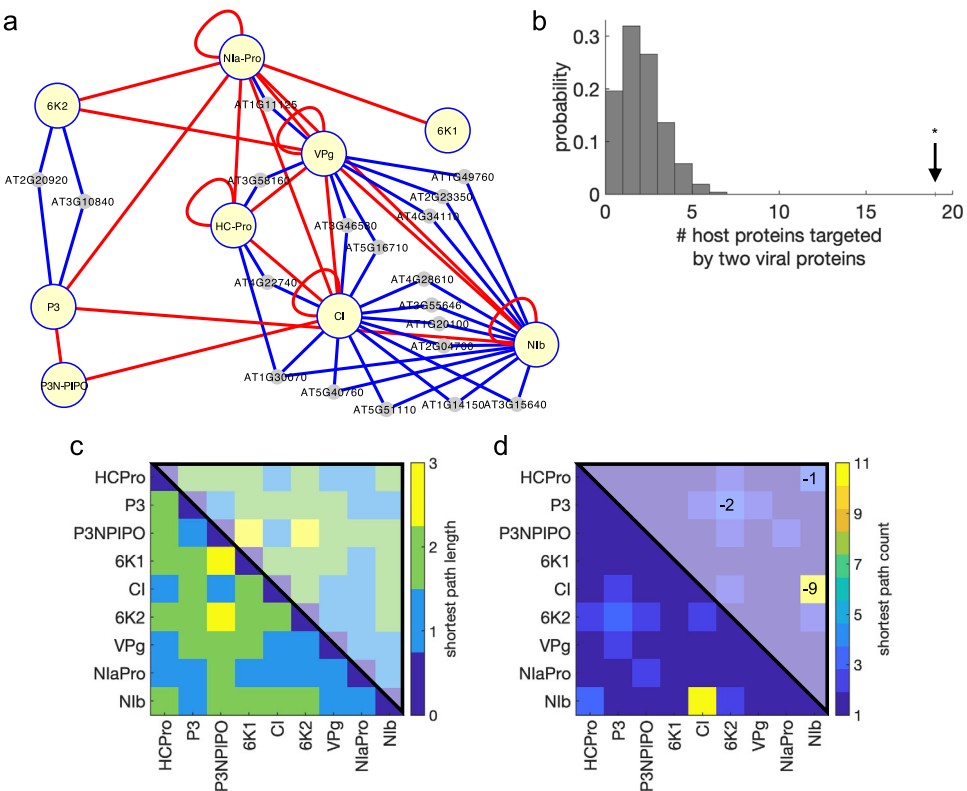

**Fig. 4 Virus–host interactions allow establishing new communication channels between the virus proteins. a** Partial virus–host interactome showing those host proteins that are targeted by two or more TuMV proteins. We refer to Supplementary Data 1 and Supplementary Data 2 for details about these 19 host proteins. **b** Null probability distribution of the number of host proteins targeted by two or more virus proteins after $10^4$ random realizations. Arrow marks the actual value; *statistical significance (z test, $P < 0.0001$). **c** Length and **d** number of the different shortest paths connecting the virus protein pairs in this partial interactome (blue and red edges). Numbers in the upper hemi-matrices indicate how these values change when only virus–virus interactions are considered (red edges).

In addition, we found a relevant negative correlation between the absolute level of differential expression upon infection and the connectivity degree (Fig. 6d). That is, the higher the connectivity degree of a given host protein, the lower the absolute differential expression upon infection, indicating that hub proteins in the AI-$1_{MAIN}$ interactome display certain robustness in expression levels in the presence of perturbations (as also observed when certain animal diseases are assessed in terms of networks)[58]. Virus targets widely distribute over this space, some being very close to its Pareto front, as observed for PLASMA MEMBRANE INTRINSIC PROTEIN 1;3 (PIP1;3), targeted by the membrane-associated P3N-PIPO; CALNEXIN 1 (CNX1) targeted by VPg; SCE1, SUMO 3 (SUMO3), HEAT SHOCK PROTEIN 70-3 (HSP70-3), the protein encoded in locus *AT1G21440* (a phosphoenolpyruvate carboxylase), and 40S RIBOSOMAL PROTEIN SA (RP40), all targeted by NIb.

Collectively, all the above results indicate that virus targets display higher expression levels than expected by chance, and also that some virus targets can be at the same time significantly up- or downregulated upon infection, provided they are not highly connected.

**Infection dynamics in plants with mutations in selected interactors.** To shed light on the role of some of the above-described interactors in the completion of the TuMV cycle, we inoculated wild-type plants and *at1g30070*, *cnx1*, *npr1-1*, *obe1*, *pcap1*, *pip1;3*, *rp40*, *sce1*, *sumo3*, and *tga1-1* mutant plants with the same amount of TuMV inocula and tracked the progression of infection for 14 days post-inoculation in terms of percentage of plants displaying systemic infection and severity of the symptoms

induced in the leaves (Fig. 7). The experiment was repeated three times for all genotypes (two in the case of *at1g30070*), always including the wild-type in each replicate. Time-course infectivity and severity of symptoms were compared in a pairwise manner with wild-type infected plants using the dynamic time warping (DTW) method. Out of the ten selected mutant genotypes, three showed significant differences for both infectivity and symptoms progression compared with wild-type: *at1g30070*, *cnx1*, and *tga1-1*. Infected *at1g30070* plants showed a significant anticipation in both curves compared to wild-type plants (Fig. 7h; harmonic mean *P*-value $HMP = 0.0008$ and $HMP = 0.0134$, respectively, with FDR correction), which suggests an antiviral role for the SGS domain-containing protein (*AT1G30070*) via interactions with HC-Pro, CI and NIb. By contrast, both curves were significantly delayed in infected *cnx1* (Fig. 7c; $HMP = 0.0015$ and $HMP = 0.0092$) and *tga1-1* (Fig. 7a; $HMP = 0.0011$ and $HMP < 0.0001$) plants, suggesting a proviral role of proteins CNX1 and TGA1, mediated by their interaction with VPg and NIb, respectively. Importantly, TGA1 has been reported as a key host factor working in the defense line against bacterial pathogens (e.g., *tga1-1* plants display increased susceptibility to *Pseudomonas syringae*)[59]. Our results reveal an opposite role of this factor when the infection is provoked by TuMV (i.e., *tga1-1* plants display increased resistance to TuMV).

The results for four other genes were dependent on the particular trait studied. Mutants *sce1* (Fig. 7i; $HMP = 0.0003$) and *pcap1* (Fig. 7e; $HMP = 0.0022$) showed a significant delay in systemic infection, but had no significant effect on symptoms curves ($HMP = 0.0907$ and $HMP = 0.0320$, respectively). This suggests that both SCE1 and PCaP1 interactions with P3N-PIPO and NIb

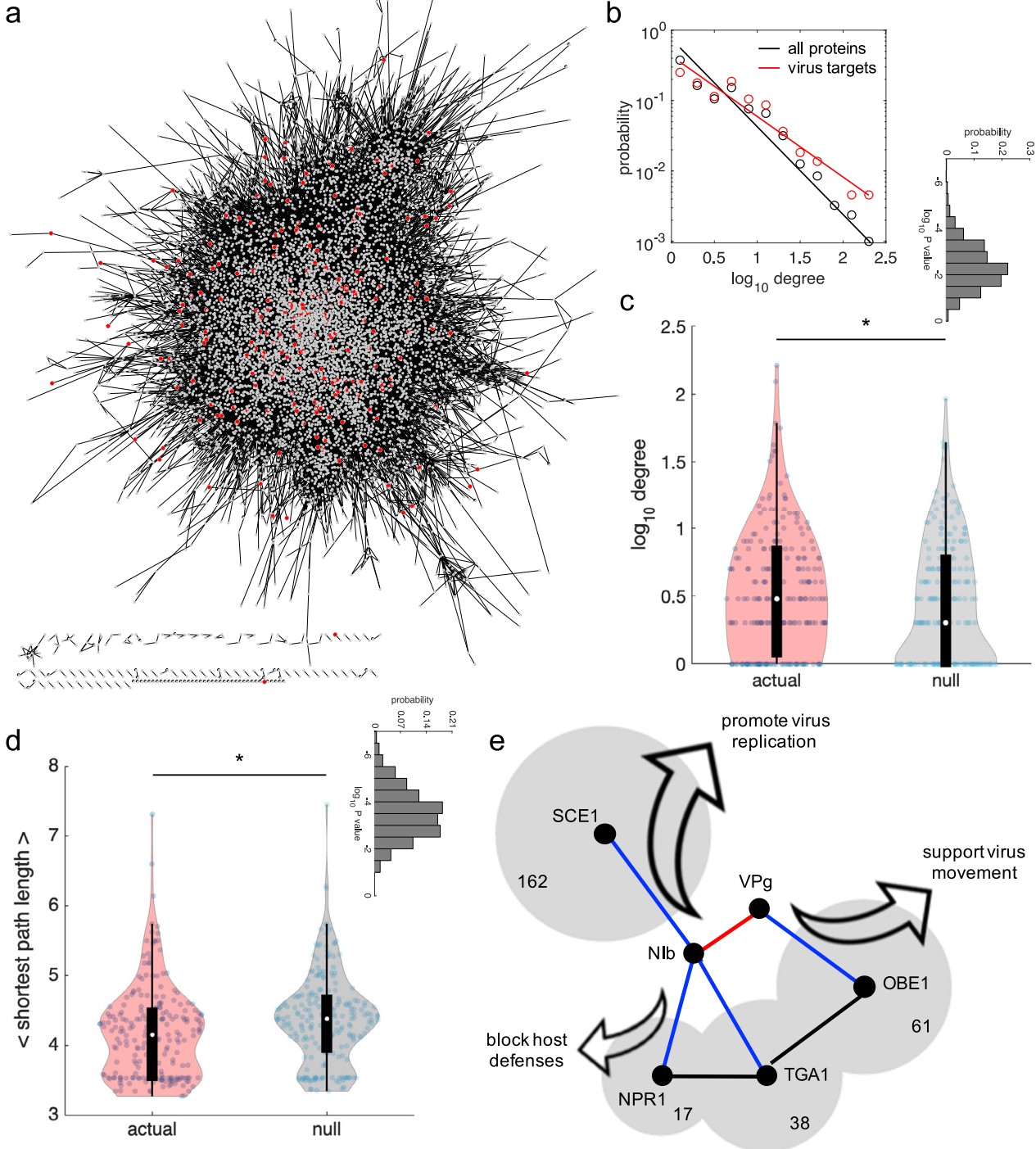

**Fig. 5 The plant virus TuMV targets host proteins with higher connectivity. a** Whole PPI of the host plant *A. thaliana* (AI-1$_{\text{MAIN}}$) contextualizing those proteins that interact with the plant virus TuMV (red nodes). **b** Probability distribution of the degree for only the virus targets (red) or all proteins in the host interactome (black). Points correspond to the data, whilst lines to the best-fitting power law probability ~ degree$^{-\gamma}$ ($\gamma = 0.860$ for virus targets, $\gamma = 1.247$ for all proteins). **c** Comparison between the actual degree distribution (from virus targets) and a representative null distribution (from randomly picked genes). *Statistical significance (Mann–Whitney *U* test, $P < 0.05$). The inset shows the distribution of *P* values after 1000 random realizations, with geometric mean 0.0045. **d** Comparison between the actual shortest path length distribution (from virus targets) and a representative null distribution (from randomly picked genes). Supplementary Fig. 3 shows the same analyses in panels **b**–**d** but excluding the 18 interactors gathered from the literature. *Statistical significance (Mann–Whitney *U* test, $P < 0.05$). The inset shows the distribution of *P* values after 1000 random realizations, with geometric mean 0.0003. **e** Network-function detail of four virus targets with markedly high degree, which interact with TuMV proteins NIb and VPg. The area of the shadow regions is log-proportional to the degree (indicated in number). TGA1 shares interactors with NPR1 and OBE1.

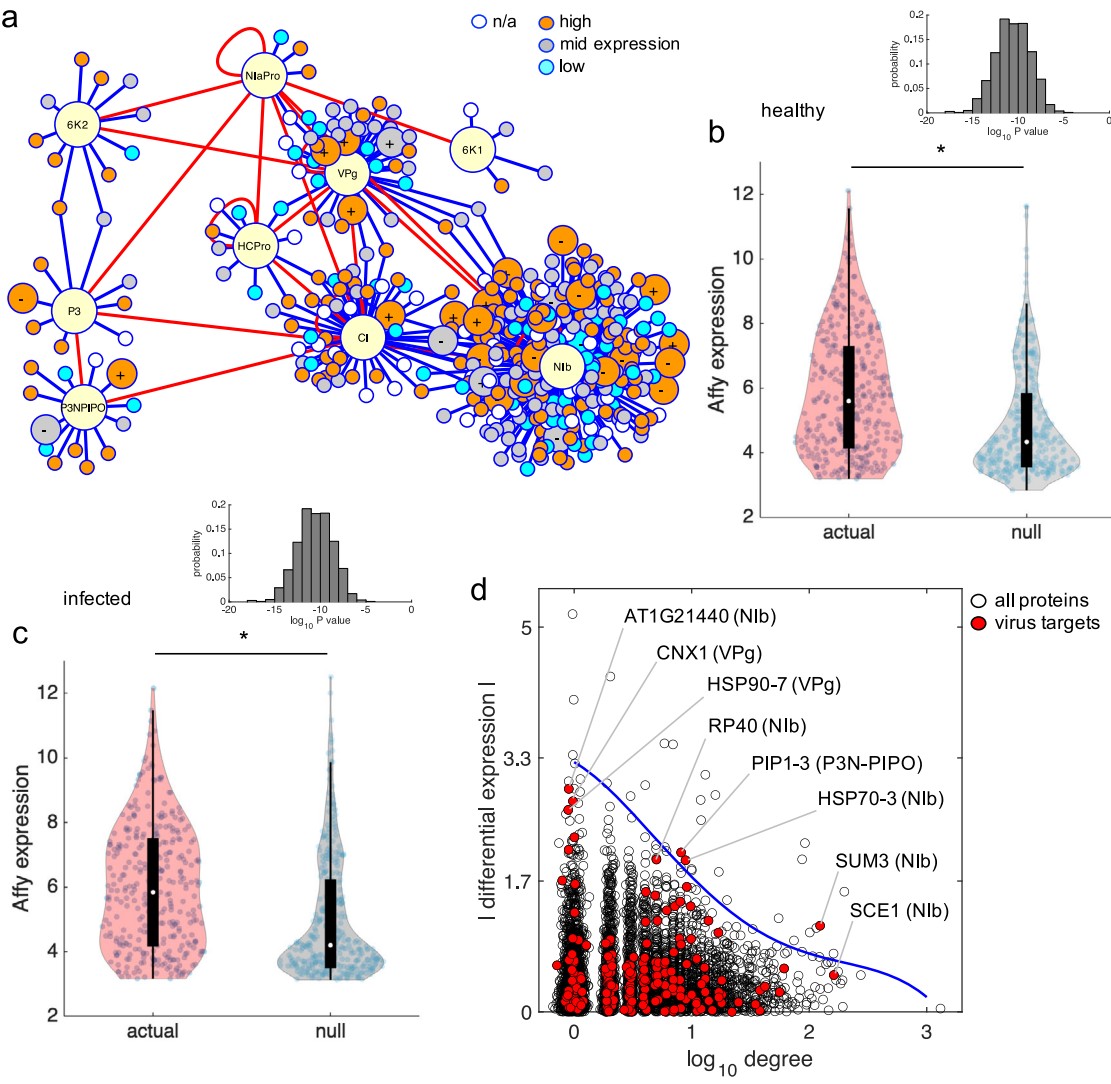

**Fig. 6 The plant virus TuMV targets host proteins with higher expression levels. a** Virus–host protein–protein interactome contextualizing gene expression data (from healthy *A. thaliana*). Three expression levels are categorized. Host proteins whose expression significantly changes upon TuMV infection are represented by bigger nodes (+indicates upregulation, −downregulation). **b**, **c** Comparison between the actual Affymetrix expression distribution (from virus targets) and a representative null distribution (from randomly picked genes). *Statistical significance (Mann–Whitney *U* test, $P < 0.05$). The inset shows the distribution of *P* values after 1000 random realizations, with geometric mean $<10^{-10}$ (in both cases). Expression levels corresponding to **b** a healthy or **c** an infected plant. **d** Scatter plot between the connectivity degree and differential expression (virus targets in red). Virus targets that are close to the soft Pareto front (blue line) are highlighted, indicating in parenthesis the corresponding virus protein.

play a proviral role in facilitating virus replication and movement without impacting the severity of symptoms. Our results agree with previous work showing that *sce1* plants show 50% reduced TuMV accumulation and weaker disease symptoms[30]. By contrast, mutants *npr1-1* (Fig. 7b; HMP = 0.0637) and *sumo3* (Fig. 7j; HMP = 0.0964) showed no significant differences with respect to wild-type plants in terms of frequency of systemically infected plants, yet showed significant delay in the progression and severity of symptoms (for *npr1-1*, Fig. 7b; HMP = 0.0073 and for *sumo3*, Fig. 7i; HMP = 0.0189). This supports the notion they may act as proviral factors that favor NIb activity during TuMV infection. Furthermore, symptoms developed by infected *npr1-1* plants were milder than those expressed by wild-type plants. No consistent effects among replicates were observed for infected *obe1* (Fig. 7d), *pip1;3* (Fig. 7f), and *rp40* (Fig. 7g) plants (i.e., HMP > FDR).

**Selective constraints upon the *A. thaliana* proteins targeted by TuMV.** Much has been discussed about the evolutionary

mechanisms that operate on the antiviral defense proteins[60]. Here, we tested the hypothesis of whether the TuMV-interacting proteins may be preferentially found in a subset of essential genes under strong purifying selection. Firstly, the within-species genome-wide ratio of nonsynonymous to synonymous polymorphisms ($p_N/p_S$) was computed from the 1001 *A. thaliana* Genomes Project[61], while the genomes of species from a sister clade, *Capsella rubella* and *Boechera retrofracta*, served as references for the analyses (Fig. 8a, results for each gene in Supplementary Data 4). We found that the average $p_N/p_S$ ratio is significantly smaller in the set of TuMV interactors than for randomly selected proteins (Fig. 8b; Mann–Whitney's *U* test, $P = 0.0016$); thus, indicating that these interactors are under strong purifying selection and particularly well conserved. As shown in previous sections, viral targets are enriched in highly connected proteins. Therefore, it is expected that if only highly connected host proteins are considered in this analysis, the above differences in average $p_N/p_S$ would become not significant. In agreement with

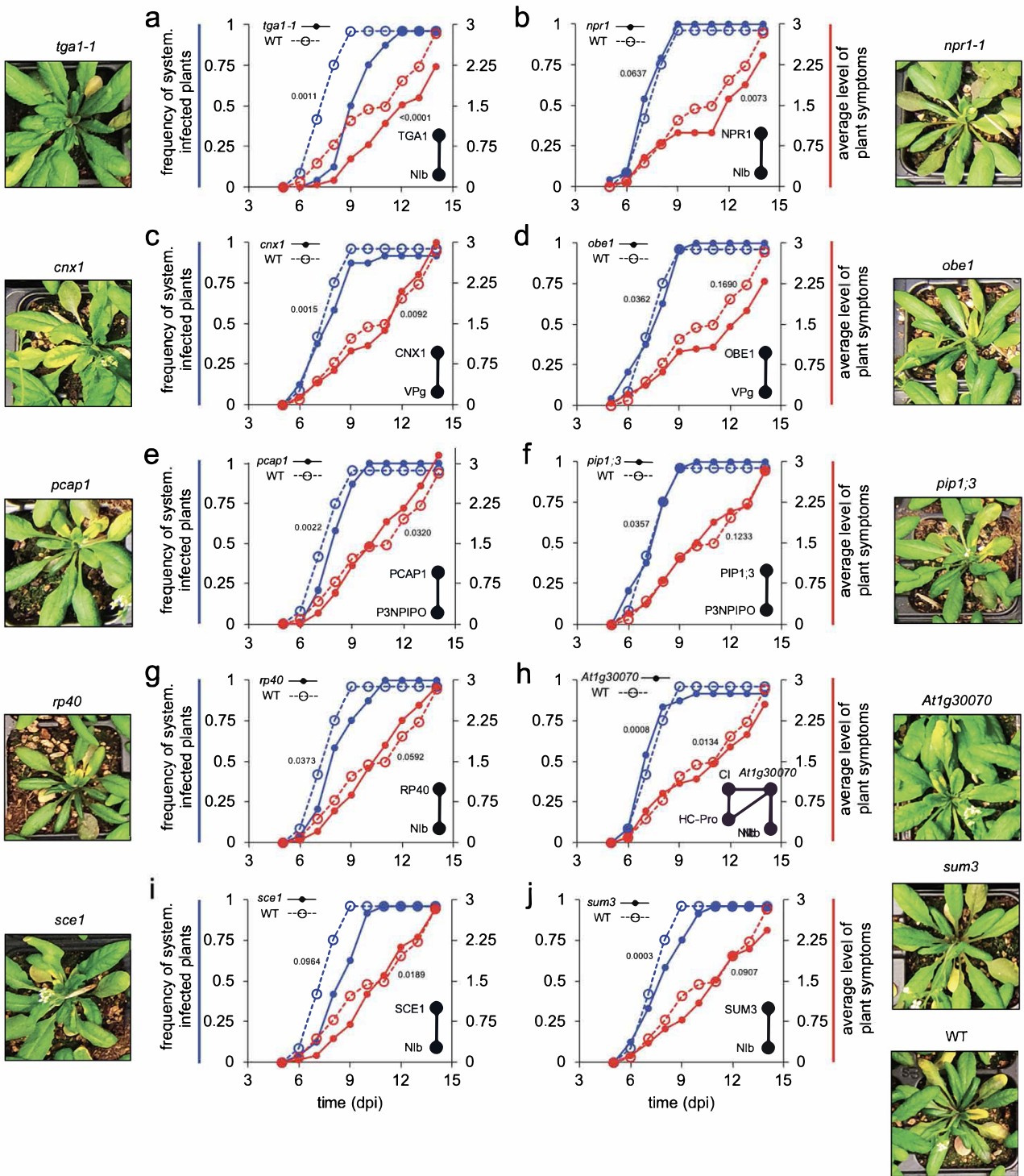

**Fig. 7 Genetic analysis and experimental validation of the effect of mutations in host proteins interacting with TuMV proteins.** Infectivity (blue) and disease progression (red) curves. As illustrative example, panels (**a**–**j**) show the results from the third experimental block comparing the increase in incidence and progression of symptoms severity for wild-type (WT) and mutant plants. Insets show the predicted interactions. Typical symptoms are indicated next to each panel. Notice that the FDR adjusted *HMP* values indicated next to each pair of curves was computed for all replicates.

this expectation, if only host proteins with degree ≥ 5 are sampled (i.e., those expected to be under strong purifying selection), no difference among viral targets and random sets exist anymore (Supplementary Fig. 4). Moreover, we estimated the proportion of adaptive nonsynonymous mutations by means of the direction of selection (*DoS*) unbiased statistic[62] in TuMV-interacting proteins and non-interacting ones in the *A. thaliana* lineage. Overall, no

significant difference exists between both groups (Fig. 8c). However, it is interesting that the proviral NIb interactor SCE1, a highly connected protein in the host, ranks second among the TuMV-interacting proteins with the largest *DoS* > 0 values (Fig. 8d). Besides SCE1, none of the other top five most evolvable TuMV-interacting proteins listed in Fig. 8d have been previously related to responses to infection. Among the most conserved

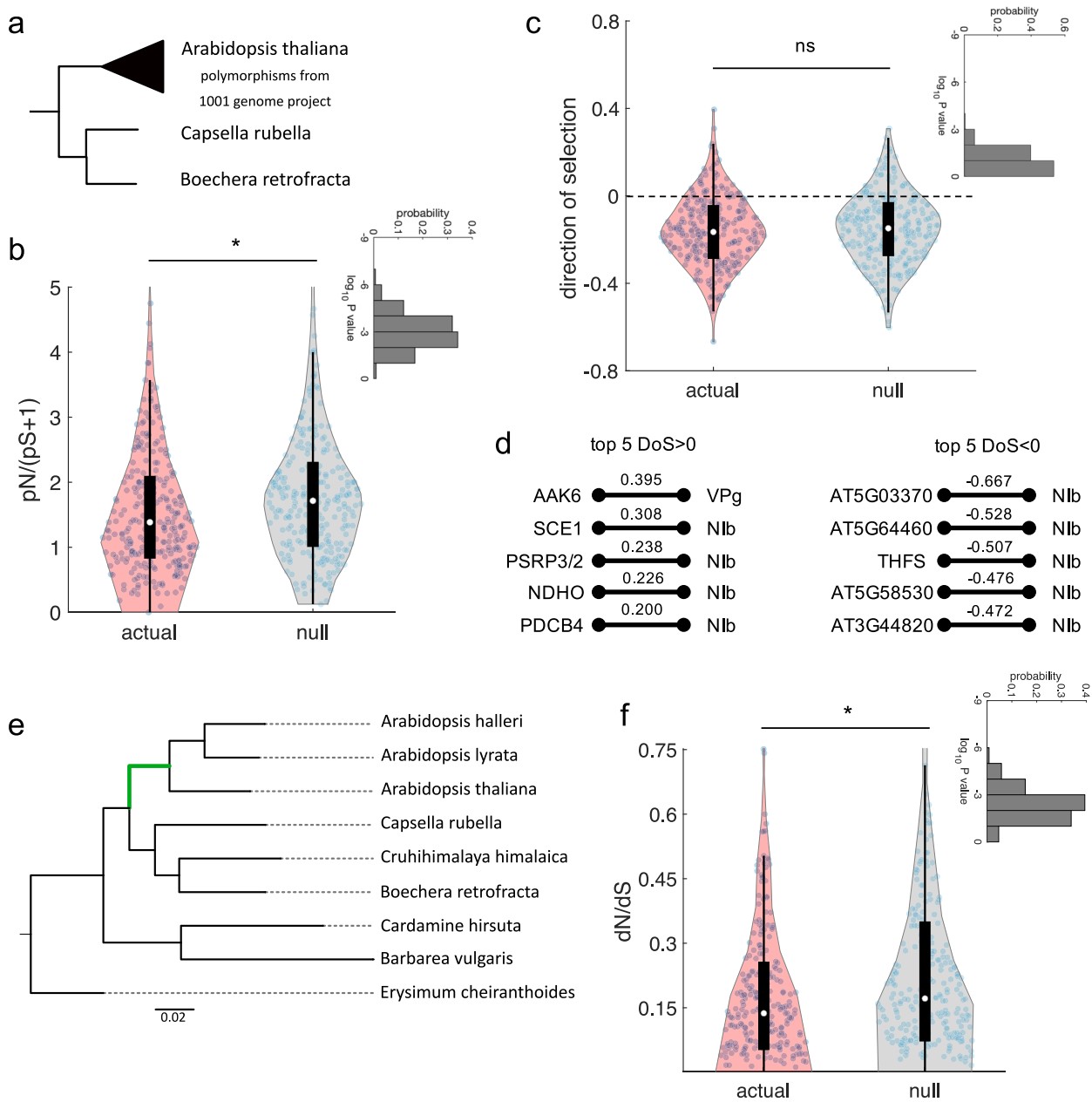

**Fig. 8 TuMV-interacting host proteins are evolutionarily conserved hubs. a** Phylogeny of the three species used for the population study. **b** Comparison between the actual $p_N/p_S$ distribution [computed as $p_N/(p_S + 1)$ to account for genes with $p_S = 0$] and a representative null distribution (from randomly picked genes). *Statistical significance (Mann–Whitney $U$ test, $P < 0.05$). The inset shows the distribution of $P$ values after 1000 random realizations, with geometric mean 0.0016. **c** Comparison between the actual $DoS$ distribution and a representative null distribution (from randomly picked genes). nsStatistically non-significant (Mann–Whitney $U$ test, $P > 0.05$). The inset shows the distribution of $P$ values after $10^3$ random realizations, with geometric mean 0.0654. **d** Top five host factors with $DoS > 0$ (positive selection) and $DoS < 0$ (negative selection), together with their interacting virus proteins. **e** Species tree generated by OrthoFinder. The branch of interest, used in the CODEML analysis, is the green branch leading to the *Arabidopsis* genus. **f** Comparison between the actual $\omega$ distribution and a representative null distribution (from randomly picked genes). *Statistical significance (Mann–Whitney $U$ test, $P < 0.05$). The inset shows the distribution of $P$ values after 1000 random realizations, with geometric mean 0.0043.

TuMV-interacting proteins ($DoS < 0$), none have been previously annotated as related to infection but with roles reported in different aspects of plant metabolism (Fig. 8d) [i.e., the acylphosphatase encoded by locus *AT5G03370*, the phosphoglycerate mutase encoded in locus *AT5G64460*, and 10-FORMYLTETRAHYDROFOLATE SYNTHETASE (THFS)]. Of note, two of the most highly connected proteins, the VPg interactor OBE1 and proviral NIb interactor TGA1 (Fig. 5e), which are also transcription factors, show $DoS < 0$ values.

Secondly, we evaluated the rates of evolution ($d_N$ and $d_S$) for each protein in the branch leading to the *Arabidopsis* genus to compute $\omega = d_N/d_S$ (Fig. 8e). We found significantly smaller $\omega$ values for the TuMV-interacting proteins than for randomly selected proteins (Fig. 8f; Mann–Whitney's $U$ test, $P = 0.0043$). This allowed us to conclude, in agreement with the results described above for $p_N/p_S$, that TuMV-interacting proteins have been subject to purifying selection and are not significantly enriched in fast-evolving genes. As in the $p_N/p_S$ analyses, if only

host proteins with degree ≥ 5 were considered for the random set, the difference between groups would disappear. In agreement with these observations, several studies have shown that fast-evolving antiviral proteins may not be representative of the many other proteins that physically interact with viruses throughout their infection cycle and that, under normal conditions, play key functions in basic cellular processes that are highjacked by the pathogens. These virus-interacting proteins usually evolve slowly in both animals[12,25,63] and plants[34,64]. Indeed, in the case of *A. thaliana*, the effectors of microbial pathogens as different as bacteria, oomycetes and ascomycetes interact with a limited set of conserved host proteins shown to be under stabilizing selection[64]. In the context of PPI networks, this observation makes extra sense as virus targets are enriched in central highly-connected hubs that likely cannot tolerate much variation without jeopardizing the functioning of the system[19].

## Discussion

**A valuable resource in plant systems virology.** A main goal of describing PPI is understanding the mechanisms by which the cell maintains its homeostasis under normal conditions and readjusts it in response to stress. Experimental virus-host interactomes are useful because they can be exploited to generate hypothesis regarding the pathogenicity and replication mode of viruses[39]. Indeed, proteins rarely act in isolation within the cell; different PPIs define major functional pathways crucial for a variety of cellular processes. In this regard, we expect that our data and analyses can inspire further studies with plant viruses. In particular, bringing together our results with some already known interactions among the different TuMV proteins and *A. thaliana* proteins (also obtained by Y2H screenings) we have been able to present a first systematic (yet surely incomplete) characterization of the PPI network between a plant virus and its natural hosts.

The *A. thaliana* protein with the largest number of interactions targeted by TuMV is the putative proviral SCE1. An emerging topic in systems virology is whether viruses exploit the host sumoylation pathway. SUMO induces proteins to change their stability, activity, and location. Multiple human DNA viruses (e.g., herpesviruses) are known to interfere with the host sumoylation pathway in diverse ways[65,66]. As previously reported, sumoylation of NIb at position K172 is a requirement for TuMV to successfully infect the plant, since *sce1* plants are resistant to infection and NIb/K172R viruses have lower infectivity[30,40].

The eleven interactors that we found for 6K2 (some shared with P3) are significantly enriched in GO terms related to vesicle transport from endoplasmic reticulum to Golgi membranes. This is interesting because all known positive-strand RNA viruses replicate their genomes in membrane-associated compartments dubbed as viral factories[52,67], which facilitate the coordination between the different factors required for RNA replication, while being protected from the RNA silencing machinery[68]. In the case of potyviruses, the small protein 6K2 is responsible for anchoring the VRC to the endoplasmic reticulum to seed the formation of viral factories[26], so the future evaluation of the effect of these host proteins on viral infection would be interesting. Interestingly, 6K2 shows strong evidence of IDPR (Supplementary Fig. 2), which may explain why this protein interacts with many different host and viral partners.

The virus-host PPI network reported here has many limitations. Firstly, TuMV proteins P1 and CP are not included in the network. We were unable to detect interactors of P1 and CP in yeast. P1 is an unstable protein only expressed as an independent polypeptide at the beginning of infection. In a previous report, protein complexes that associate with P1 *in planta* were retrieved by following an alternative strategy of affinity purification

coupled to mass spectrometry[69]. By contrast, CP is the coat protein of the virus and may display a self-binding ability in yeast that precludes the interaction with other proteins[70]. Though this possibility exists, it might not be the only plausible explanation; other virus proteins, such VPg, also self-assemble and we have been able to identify candidate interactors. Furthermore, our study only considers the mature proteins of the virus, but the polyprotein by itself (e.g., including P1) and the viral RNA can establish specific interactions with the host, which should be integrated into a larger network. Secondly, some binary interactions already reported between TuMV and *A. thaliana* escaped our HT-Y2H screening[26]. Other interactions that were previously captured by affinity purification and that we did not obtain, such as between CP and the chaperon HSP70 and its cochaperone CPIP[71], might be rationalized either as interactions through third parties or as false negatives. Moreover, the AI-1_MAIN interactome is still incomplete as it only covers ~8000 proteins (about one-third of the total proteome)[54]. Consequently, several host proteins identified here to interact with the virus are not included in the computational network analysis. Thirdly, virus proteins usually perform multiple functions at different stages of the infection cycle[26]; thus, the virus-host interactome is surely time-dependent. Interactions can also change if the virus accumulates nonsynonymous mutations that affect the binding interface of the virus proteins. In this regard, our results only provide a snapshot of a highly dynamic process[72]. Hence, further experimental and computational work is required to complete the picture for TuMV and other potyviruses to assess differences and commonalities in their mode of action.

**Linking the TuMV-*A. thaliana* interaction map with disease etiology.** Plants and their viruses are engaged in an arms race[60]. On one hand, pathogens deploy virulence effectors into host cells, wherein they establish highly dynamic physical interactions with host proteins to redirect the cellular resources for their own benefit[72,73]. On the other hand, plants sense the presence of pathogens at two different levels. Firstly, plants recognize conserved pathogen-associated molecular patterns (PAMPs) by pathogen-recognition receptors exposed on the outer side of the cell membrane. This first level of recognition results in PAMP-triggered immunity (PTI)[73,74]. Secondly, plants deploy a set of intracellular immune receptors of the nucleotide-binding site leucine-rich repeat protein family, analogous to the animal innate immune NOD-like receptors. This activation results in effector-triggered immunity (ETI), which amplifies PTI responses, resulting in a burst of reactive oxygen species, changes in ion fluxes, increases in cytosolic $Ca^{2+}$ levels, activation of mitogen- and $Ca^{2+}$-dependent protein kinases, elevation of phytohormones (most remarkably SA), and transcriptional reprogramming, often leading to host cell death (hypersensitive response) and long-lasting systemic acquired resistance (SAR)[73].

We found that NPR1 interacts with the virus replicase NIb. NPR1 is an essential positive regulator of SA-induced pathogenesis-related (*PR*) genes expression and pathogen induced systemic resistance. Moreover, SAR and *npr1* mutants usually show increased enhanced susceptibility to bacterial biotrophs[75]. However, our results are suggestive of a different role during viral infections, as *npr1-1* plants show delayed and weaker symptoms. In the nucleus, NPR1 interacts with TGA bZIP transcription factors, which also control the expression of *PR* genes[59]. Furthermore, NPR1 plays a role in histone modification, enforcing the priming of SA-induced defense genes and transgenerational immune memory[76]. Related to this point, SUMO3 plays an essential role in regulating the function of NPR1[77]. It has been reported that infection with TuMV results in

upregulation of *SUMO3* expression and the interaction between SUMO3 and NIb is necessary for successful infection[31]. In fact, sumoylation of NIb in the nucleus counteracts the SUMO3-activation of NPR1[31], resulting in the blockage of the PR-mediated resistance pathway. Sumoylated NIb is translocated from the nucleus to the cytoplasm to form the VRC. Our results suggest that this process might be mediated by a binary interaction between NIb with NPR1 that likely takes place in the nucleus, affecting both expression of *PR* genes as well as NIb trafficking to the cytoplasm and resulting in milder symptoms.

Interestingly, NIb also interacts with the strong proviral TGA1 transcription factor (a hub in the host proteome), which is presumed to enhance the negative effect on SA-mediated resistance of the aforementioned interaction, especially because this targeting will produce a feedback response in the system by downregulating the production of SA[78].

Another essential component of the *A. thaliana* immune response affected by TuMV infection is the production of reactive oxygen species and $H_2O_2$ influx[79,80]. In particular, $H_2O_2$ is produced in the apoplast by plasma membrane-associated NADPH oxidases RESPIRATORY BURST OXIDASE HOMO-LOGS (RBOH). Then, $H_2O_2$ is translocated into the cytoplasm by several aquaporins[81]. Once there, it cross-talks with PTI and SAR pathways via redox conformational changes of NPR1 and activation of a mitogen-activated protein kinase cascade that upregulates a set of immune responses, including subsequent production of $H_2O_2$ and callose deposition[81]. Mutations in *RBOHD* and *RBOHF* genes result in a reduction in $H_2O_2$ influx but in enhanced resistance to TuMV[80]. Interestingly, we found the membrane-associated P3N-PIPO to interact with the aquaporin PIP1;3. If this interaction modifies the influx of $H_2O_2$, then one should expect an alteration in the susceptibility of *pip1;3* plants to TuMV infection. However, we found no significant effect of *pip1;3* mutation in disease progression, suggesting that the interaction between PIP1;3 and P3N-PIPO is not relevant in terms of virus infection.

In addition, we took advantage of the virus-host interaction map to gain mechanistic insight about one of the most remarkable symptoms of TuMV infection, i.e., the sterilization of *A. thaliana* plants. This results from the arrested apical meristem growth[38]. VPg has been described as a scaffolding protein that interacts with other viral proteins as well as with many host proteins, most notably canonical factors involved in protein synthesis[82]. Interestingly, even though we found no effect on disease progression, the interaction between VPg and OBE1 (an evolutionarily conserved hub in the host proteome) is suggestive of a putative new role of VPg in pathogenesis, since OBE1 is involved in the establishment and maintenance of the shoot and root apical meristems through the regulation of the expression of the stem cell factor WUSCHEL (WUS). It is interesting to note that VPg also interacts with the PDH fingers proteins OBERON 2 (OBE2), an element that functions like OBE1[83], and OBERON 3 (OBE3), one of the transcriptional activators of WUS and the CLAVATA (CLV) pathway[84]. We did not observe an effect of mutating *OBE1* in disease progression, which might suggest the existence of functional redundancy with the product of *OBE2* and *OBE3*. By looking at the interactors of OBE1 in the network, we found OBE3, STOMATAL CLOSURE-RELATED ACTIN BINDING 1 (SCAB1), which stabilizes actin filaments and controls stomatal movement[85], and various WRKY transcription factors, which activate the expression of defense genes against pathogens[86]. But as recently shown, WUS is also responsible for triggering innate antiviral immunity in the meristems[87]. Consequently, we may argue that OBE1 and OBE3 are factors sequestered by TuMV in order to facilitate its systemic movement[28] by subverting the regulation of both the

stomatal mechanics and the antiviral response in stem cells, with the side effect of sterilization. However, further experiments are required to confirm this hypothesis, since, as stated, *obe1* mutant plants did not show a significant effect on infection.

**Comparing the mode of action of TuMV and human viruses**. As discussed above, TuMV interferes with PTI and ETI. Mammalian cells also trigger their innate immune systems by PAMP recognition through two groups of receptors, the Toll-like and RIG-I-like receptors[88]. Similar to what occurs in plants, these receptors initiate a signaling cascade that converge on transcription factors that induce type I interferon (IFN-α/β) expression. Not surprisingly, mammalian viruses also block the IFN-α/β pathway; the V protein of measles virus is a well-studied example[89].

From a holistic perspective, some principles have been put forward to characterize virus-host interactions in humans. Broadly, viruses tend to interact with proteins that are (i) hubs and bottlenecks in the host interactome[9,14], (ii) evolutionarily conserved and under positive selection[24,25], (iii) involved in key biological processes instrumental for their infection and replication[14], and (iv) close to other proteins involved in disease symptoms[23]. Our results demonstrate that these principles generally hold true in the case of a plant virus, while adding some nuances.

Firstly, we show that the set of TuMV partners is significantly enriched in hub proteins, which suggests that the perturbation introduced by the virus is not merely random. This pattern has been described for several human viruses, including both RNA (e.g., hepatitis C virus or dengue virus) and DNA (e.g., Epstein-Barr virus) viruses[54]. This way, the virus has the potential of dismantling the entire system through a selective attack[21], which aligns with the so-called centrality-lethality rule[19]. Moreover, by leaning on host hub proteins, a small set of viral proteins can lead to the simultaneous rewiring of many cellular processes. However, hubs only represent a small fraction of all nodes in a scale-free network, so this proportion is also transmitted to the virus targets. Our results also reveal that TuMV selectively targets proteins that, while not being highly connected, bridge different parts of the *A. thaliana* interactome, thereby with increased ability to spread information (i.e., proteins that have high neighborhood connectivity)[22,35]. In addition, the virus protein by itself may work as a bridge to coordinate the perturbation among different host factors, as may be the case of HC-Pro, P3N-PIPO or NIb by targeting proteins with very different functions in the cell.

Secondly, by analyzing both the *DoS* and *ω* ratios, we found that the TuMV targets are significantly enriched in proteins whose genes have evolved slowly. This is in tune with previous analyses of human genes targeted by virus factors[12,25,63] and suggests that virus proteins preferentially interact with evolutionarily constrained elements with the aim of ensuring broad host ranges. We then evaluated if even at slow rates these virus targets have been subject to positive selection, finding non-significant results. This is in contrast with the concept of an evolutionary arms race, as well as with previous analyses indicating that virus targets in humans have been adapted in response to infection[25]. Perhaps, a new evolutionary study restricted to viral protein binding surfaces rather than whole gene sequences would offer more enlightening results[24]. We also found that the TuMV targets display higher expression levels, which indeed favors the interaction with multiple proteins in different tissues. Interestingly, it is well accepted that highly expressed genes tend to evolve at slower rates[90], arguably because the exploration of new variants is more costly[91]. Thus, conservation and expression are two sides of the same coin that the virus conveniently exploits.

Thirdly, our functional analysis showed that the host proteins targeted by the virus are significantly enriched in GO terms associated with defense and regulation, but also with general metabolic processes. Although it is difficult to establish a frontier, the study of the mode of action of multiple human viruses revealed that proteostasis, signaling (e.g., JAK-STAT pathway, which acts downstream of IFN-α/β), transport, and RNA metabolism are host functions preferentially targeted by RNA viruses, while transcription, proteostasis, macromolecular assembly, DNA and RNA metabolism, and cell cycle (e.g., cancer pathways) are the affected by DNA viruses[14]. TuMV, a plant RNA virus, seems to be closer to human RNA viruses, especially because it impairs hormone signaling pathways, host protein expression at the level of translation, and the RNA silencing machinery.

Fourthly, it has been argued that proteins involved in disease susceptibility and symptomatology in humans (e.g., cancer-related proteins altered by viral infections, like Epstein-Barr virus-associated lymphoma) should reside in the network neighborhood of the corresponding viral targets[16,23]. In this regard, the link established between VPg, OBE1 (virus target), WUS (host protein regulated by the virus target), and sterilization (disease) in our phytopathosystem is a good example. All in all, we anticipate gaining a deeper understanding about the mode of action of TuMV and other plant viruses if the interrelation interface is enlarged with more interactions (e.g., by curating and integrating different sources of information)[39] and also if the *A. thaliana* interactome is upgraded, which now contains ca. 22,000 interactions. In addition, further work should be aimed at combining PPIs, transcriptional regulation, and metabolic pathways to generate a mechanistic picture of viral infection in plants as comprehensive as possible[7].

## Methods

**Plasmid construction**. The plasmid p35STunos contains an infectious cDNA clone (GeneBank accession AF530055.2) corresponding to the TuMV isolate YC5 obtained from infected calla lily (*Zantedeschia* sp.)[92]. The 11 TuMV cistrons (corresponding to the virus proteins P1, HC-Pro, P3, P3N-PIPO, 6K1, CI, 6K2, VPg, NIa-Pro, NIb, and CP) were amplified by polymerase chain reaction (PCR) from p35STunos with the Phusion High-Fidelity DNA polymerase (Thermo) by using the corresponding pairs of primers, including Gateway adapters, listed in Supplementary Data 5.

For the Y2H system based on the *GAL4* promoter[36], the PCR products were cloned by recombination with the In-Fusion enzyme (Clontech) into the yeast bait vector pGBKT7 (Clontech), which was digested with *Eco*RI and *Bam*HI. This generated a translational fusion of the virus protein (bait protein) with the GAL4 DNA-binding domain. The construction for P3N-PIPO was done in two steps. First, part of the P3 cistron was amplified by PCR and cloned into the plasmid pGBKT7. Second, one adenine was inserted in the putative frameshift site (GGAAAAAA) by site-directed mutagenesis to express the virus protein without the need of frameshifting[93].

For the screening based on the sUbq (interactions occurring in the membrane)[40], the PCR products were cloned by recombination in vivo to obtain the CubPLV translational fusion (in the case of P3, P3N-PIPO, and 6K2). For that, the bait vector pMetYC-gate was digested with *Pst*I and *Hin*dIII and then was co-transformed together with the PCR product into the yeast strain THY.AP4. Transformants were selected on SD/-Leu medium after incubation at 30 °C for 5 days.

For the BiFC constructs, PCR products from all TuMV genes were recombined into the plasmid pDONR207 by using the BP Clonase II Enzyme mix (Invitrogen). For cloning the different *A. thaliana* Col-0 genes, total RNA was extracted from plant tissues by using the Trizol reagent (Invitrogen) following the manufacturer's recommendations and further purified by LiCl precipitation. The corresponding cDNAs were synthesized by using the RevertAid H Minus First Strand cDNA synthesis kit (Fermentas) with a polyT+N-primer. Full-length ORFs were amplified by PCR from those cDNAs with the Phusion High-Fidelity DNA polymerase (Thermo) by using suitable primers, including Gateway adapters (Supplementary Data 5). The constructs were then recombined into the plasmid pDONR207 by using the LR Clonase II Enzyme mix (Invitrogen). All constructed plasmids were amplified in *Escherichia coli* strain DH5α, purified, and verified by sequencing.

**HT-Y2H screening**. To identify the host proteins that interact with the eleven TuMV proteins, an *A. thaliana* Col-0 cDNA library (Clontech) was screened by using the Matchmaker Gold Yeast Two-Hybrid System (Clontech). For this, the Y187 haploid yeast strain (with library proteins) and the Y2HGold haploid reporter strain (with the plasmid pGBKT7 that expresses each of the 11 TuMV proteins) were mated and plated on a double dropout medium (SD/-Leu/-Trp) containing 40 μg/mL X-α-Gal and 200 ng/mL aureobasidin A, and then incubated at 30 °C for 5 days. Co-transformants that were phenotypically positive for α-galactosidase activity were subjected to a further, more stringent phenotypic assay on a quadruple dropout medium (SD/-Leu/-Trp/-Ade/-His) containing X-α-Gal and aureobasidin A. Plasmids pGBKT7-T-antigen, pGADT7-laminin C, and pGADT7-murine p53 (Clontech) were used as negative controls. Colony PCR were performed from the yeast colonies that displayed a positive interaction to eliminate duplicate clones, and prey plasmids were rescued with an isolation kit to be subsequently transformed into *E. coli* DH5α for amplification and sequencing, as described by the manufacturer. DNA and protein sequence analyses were performed with the WU-BLAST algorithm as formerly implemented in the TAIR website[94]. For each novel interaction, the prey and bait plasmids were co-transformed into the Y2HGold strain to verify genuine positive interactions and transformed on their own to remove false positives.

According to the manufacturer's information, the plant cDNA was normalized prior to library construction to reduce the copy number of abundant cDNAs derived from highly represented mRNAs, thereby increasing the representation of low copy number transcripts. Therefore, identified interactors should not be biased towards highly expressed genes. The library contains $1.1 \times 10^7$ independent clones of ~1.30 kb of size (range 0.7–3.0 kb), which given the ca. 27,416 protein coding genes in *A. thaliana* genome (TAIR release 10), and ignoring isoforms encoded by alternative transcripts, provide an average redundancy of ca. 400 clones per mRNA.

An *A. thaliana* Col-0 cDNA library based on the sUbq system (Dualsystems) was also screened to identify membrane-associated interactions. For this, the yeast THY.AP4 (already with the plasmid pMetYC-gate that expresses each of the four membrane-associated TuMV proteins) was transformed with the library proteins fused to the NubG domain, plated on a triple dropout medium (SD/-Leu/-Trp/-Ade), and incubated at 30 °C for 10 days. Methionine at 10 μg/mL was added to the medium to optimize the CubPLV fusion expression. Co-transformants were subjected to further, more stringent phenotypic assay on a quadruple dropout medium (SD/-Leu/-Trp/-Ade/-His) containing 80 μg/mL X-Gal and 10 μg/mL methionine. Colony PCR were performed from the yeast colonies that displayed a positive phenotype for β-galactosidase activity to eliminate duplicate clones, and prey plasmids pDSL-Nx were rescued in *E. coli* DH5α for sequencing. The pMetYC-gate Ost3 plasmid (Dualsystems) served as a negative control. False positives were eliminated by testing for autoactivation.

The prevalence of IDPR as a plausible explanation for the observed number of interactors per TuMV protein was evaluated using the deep neural network algorithm described in ref. [46] and available online at the fIDPnn server (http://biomine.cs.vcu.edu/servers/fIDPnn). The output of the algorithm is a disorder propensity score per amino acid residue.

**BiFC assay with confocal microscopy**. The Gateway destination vectors pYFN43 and pYFC43 (kindly provided by Prof. Pablo Vera, IBMCP) were used to obtain the coding sequences of the two moieties of a yellow fluorescent protein (YFP)[95]. The primers used to amplify by PCR the N-terminal sequence of YFP (corresponding to residues 1 to 154) and the C-terminal sequence of YFP (corresponding to residues 155 to 240) are also listed in Supplementary Data 5. The Phusion High-Fidelity DNA (Thermo) was used. Both PCR products were cloned into the plasmid pEarlyGate101 with the restriction enzymes *Avr*II and *Spe*I, replacing the native YFP cDNA. The Gateway destination vectors p101-YFN and p101-YFC were created in this work to generate the translational fusions with the virus and host proteins.

Cultures of *Agrobacterium tumefaciens* strain C58 harboring appropriate binary plasmids were grown overnight and then centrifuged, and $OD_{600}$ was adjusted to 0.5 with 10 mM MES pH 5.6, 10 mM $MgCl_2$, and 150 mM acetosyringone. Individual bacterial cultures were mixed and used to agroinfiltrate the young leaves of 2–3 weeks old *N. benthamiana* plants (for the simultaneous transient expression of the two YFP moieties fused to the proteins of interest). After 48 h, the yellow fluorescence of agroinfiltrated leaves was analyzed by using an inverted confocal microscopy (Zeiss LSM780) with a CApo 40×/1.2 objective (Zeiss). To detect yellow fluorescence (from reconstituted YFP), excitation was done with a 488 nm argon laser and the resulting emission signal was collected in the 520–550 nm window; while to detect red fluorescence (from chloroplasts), excitation was done at 488 nm and emission collected at 680–750 nm. Image processing was performed with ImageJ v1.8[96].

**Experimental validation of interactors using knock-out mutant lines**. Seeds of the *A. thaliana* EMS mutant *npr1-1* and of the T-DNA insertion lines were obtained from Nottingham Arabidopsis Stock Center (NASC), except *tga1-1*, kindly provided by Prof. Christiane Gatz (Georg-August University, Göttingen, Germany). All *A. thaliana* mutant lines were in the Col-0 genetic background (Supplementary Table 1). For the selection of homozygous plants of segregating lines, genomic DNA was isolated using hexadecyltrimethylammonium bromide

and amplified by PCR using two gene-specific oligonucleotides flanking the location of the T-DNA insertion and one additional oligonucleotide specific for the T-DNA. Oligonucleotides were designed using the iSect Primers Tool from the Salk Institute Genomic Analysis Laboratory (SIGnAL, http://signal.salk.edu/tdnaprimers.2.html) (Supplementary Table 2).

The TuMV isolate YC5 described above was used as inoculum for plant infection. The virus was maintained in *N. benthamiana* plants. For inoculation of *A. thaliana* plants, finely ground infected leaf material from *N. benthamiana* was suspended at 100 mg/mL with inoculation buffer [50 mM phosphate buffer (pH 7), 3% polyethylene glycol (PEG) 6000, and 10% carborundum at 100 mg/mL (diluted in the same PEG/phosphate buffer)]. Two leaves of 24 *A. thaliana* plants (3 weeks old) were mechanically inoculated with 5 μL of the prepared suspension. As a control, another set of plants were mock-inoculated using only inoculation buffer. This experiment was done in triplicated. Each plant was daily monitored for symptom development for 14 days. A plant was diagnosed as infected when it showed symptoms of grade 1 according to the scale described in ref. [97]. In addition, to evaluate the progression of disease severity for different plant genotypes, each day the symptoms of each cohort of 24 plants were averaged.

Infectivity and symptoms progression curves for mutant and wild-type infected plants were compared in a pairwise manner using the DTW algorithm that measures the similarity between two temporal sequences[98] as implemented in the R package dtw version 1.22–3. The statistical significance of the estimated distance value was evaluated by bootstrapping, with replacement, the values of both curves at the same time points (10,000 pseudosamples). The bootstrap *P* values generated for each experimental replicate were collapsed into a single significance test using the harmonic mean *P*-value (*HMP*) method for combining significance values from independent tests of the same null hypothesis[99]. Finally, Benjamini–Hochberg FDR procedure was used to adjust the *P* values among plant genotypes[100]. These analyses were done with R version 4.1.2 under RStudio version 2022.02.3.

**Computational functional analysis**. With the whole list of TuMV-targeted host proteins (all virus proteins), a functional analysis was performed by using the agriGO webserver[101] to identify which gene ontology categories (related to biological processes) are over-represented. The statistical significance, with respect to the complete plant genome (TAIR release 10), was evaluated by a Fisher exact test ($2 \times 2$ contingency table) with a correction for multiple testing using the FDR, only considering GO terms with five or more mapping entries. With those identified biological processes, a functional network in a semantic space was constructed by using the REVIGO tool[102]. This analysis also served to identify the particular biological processes in which the targets of a given virus protein participate. A map between the different virus proteins and 112 relevant biological processes (corresponding to metabolism, development, organization, signaling, regulation, and defense) was generated with the fraction of virus targets implicated in each process. Note that a given host protein can participate in multiple processes.

**Autogenous host and virus interactomes**. The *A. thaliana* PPI network model AI-1$_{MAIN}$ was constructed by accounting for all binary interactions with experimental evidence (mainly by Y2H screens)[54,103]. This network covers ~8,000 plant proteins and has ~21,600 non-redundant interactions between them (Supplementary Data 3). AI-1$_{MAIN}$ was used to contextualize the host proteins identified as targets for the different virus proteins. A pre-analysis of the global topological properties of AI-1$_{MAIN}$ was performed with Cytoscape[104]. This quantitative evaluation included the computation of the connectivity degree and average shortest path length for each node in the network. The whole *A. thaliana* interactome as well as the TuMV-*A. thaliana* interactome here identified were also represented with Cytoscape. In addition, a general PPI of potyviruses was taken from previous work that collected experimental data from multiple Y2H and BiFC assays[44]. The recently described interaction between P3 and P3N-PIPO was also included[105].

**Network analysis**. The degree distribution (for AI-1$_{MAIN}$ interactome) was represented for the TuMV-targeted proteins and for all plant proteins, fitted in both cases to a probability power law: $P(k) \sim k^{-\gamma}$, where $k$ is the connectivity degree and $\gamma$ the scale coefficient[20], done with MATLAB (MathWorks). In addition, for the list of TuMV-interacting proteins, the observed distributions of connectivity degree, average shortest path length (topological properties), expression levels (in healthy and infected states), rates of gene evolution, and genome-wide ratios of nonsynonymous to synonymous polymorphisms (see below) were represented as violin plots. One thousand random lists of proteins were also generated to compute null distributions of these variables. The statistical significance was assessed by means of Mann–Whitney *U* tests with MATLAB.

**Gene expression data**. Transcriptomic data of healthy and TuMV-infected *A. thaliana* plants (corresponding to Affymetrix data from microarray experiments) were retrieved from previous work[57], which were subsequently normalized by state-of-the-art procedures in a meta-analysis that studied multiple plant viruses[106] to obtain absolute and relative (infected vs. healthy) expression values. In addition, a scatter plot between differential expression and connectivity degree was generated for all host proteins, highlighting the virus targets. A soft Pareto front providing an approximation to the optimal expression and connectivity degree levels was

computed with MATLAB (1000 different strict Pareto fronts were computed from samples of 10% of all proteins, bootstrapping, and the average front was then computed).

**Comparative genomics and quantifying adaptation**. To determine the evolutionary rate leading to the *Arabidopsis* genus, we selected three *Arabidopsis* species (*Arabidopsis halleri*, *Arabidopsis lyrata* and *A. thaliana*), three species from a sister clade (*B. retrofracta*, *C. rubella* and *Crucihimalaya himalaica*) and three representations of an outgroup (*Cardamine hirsuta*, *Erysimum cheiranthoides* and *Barbarea vulgaris*; Supplementary Table 3). Only long splicing variants were kept to minimize the complexity of the dataset. Furthermore, proteins shorter than 50 amino acids were filtered out. Orthologous gene groups were detected with OrthoFinder[107], which was run with default setting on the nine genomes. The orthologous groups were pruned for duplicates by only keeping the best BLAST hit against the *A. thaliana* gene copy. Further increment of the dataset was done by allowing a certain amount of absence in some of the genomes, with the absence being allowed if the closest relative was present (Supplementary Fig. 5). Amino acid alignments were generated with MAFFT[108] using the accurate option (L-INS-i). Regions containing a large number of gaps were removed from the back-translated codon alignments using trimAL[109] with 0.85 gap-score cut-off. The alignment dataset contains a variable number of taxa, meaning that CODEML from the PAML package[110] could not be run directly. GWideCodeML[111] overcomes this problem by pruning the given species tree to fit each of the alignments. The $\omega = d_N/d_S$ heterogeneous branch model of CODEML was run with default setting on GWideCodeML providing the unrooted species tree generated by OrthoFinder.

The genome set was reduced for the population study to containing *A. thaliana* and two close relatives (*B. retrofracta* and *C. rubella*). The information from the 1001 *A. thaliana* Genomes Project[61] was used for generating artificial *A. thaliana* genomes contains all SNP variants observed. Orthologous groups obtained in the previous analysis was used again here. Amino acid alignments were done as previously specified with MAFFT and trimAL. Determining the proportion of adaptive amino acid substitutions within *the A. thaliana* population was done using the direction of selection unbiased statistic (*DoS*)[62]. *DoS* calculates the difference between the proportion of substitutions and polymorphisms that are nonsynonymous as $\mathrm{DoS} = D_N/(D_N + D_S) - p_N/(p_N + p_S)$, where $D_S$ and $D_N$ are the numbers of fixed and $p_S$ and $p_N$ are the numbers of polymorphic mutations per gene within the *A. thaliana* population (subindexes $S$ and $N$ refer, respectively, to synonymous and nonsynonymous mutations). *DoS* takes values in the interval $[-1, 1]$ and under the null hypothesis of neutral evolution $DoS = 0$; adaptive evolution would result in $DoS > 0$, while $DoS < 0$ values are expected for purifying selection. The R library PopGenome version 2.7.5[112] was used for the calculation of $D_N$, $D_S$, $p_N$, and $p_S$. The results of all the above analyses are presented in Supplementary Data 4.

**Statistics and reproducibility**. Statistical tests are mentioned throughout the text as presented. In all cases, contrast statistic and significance level are provided.

**Reporting summary**. Further information on research design is available in the Nature Portfolio Reporting Summary linked to this article.

## Data availability
All relevant data are included in the files of the publication (main figures and supplementary material). Any other data that support the findings of this study are available from the corresponding authors upon request.

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

## Acknowledgements

We thank Francisca de la Iglesia and Paula Agudo for excellent technical assistance and the rest of the EvolSysVir lab for fruitful discussions. This work was supported by grants PID2019-103998GB-I00 and PGC2018-101410-B-I00 (Agencia Estatal de Investigación - FEDER) to S.F.E. and G.R., respectively, and PROMETEO2019/012 (Generalitat Valenciana) to S.F.E.

## Author contributions

S.F.E. conceived the research with input of G.R.; F.M. performed the experimental work (HT-Y2H screens and BiFC assays), supervised by L.Y. (partly) and S.F.E.; J.L.C. performed the experimental work (mutant analyses); G.R. performed the computational work (functional, network, and evolutionary analyses) and prepared the figures, with input of C.T. (evolutionary analysis) and S.G-S. (literature curation); J.H. worked in the intravirus interaction network; C.T. performed the comparative evolutionary analyses; L.Y. contributed with materials (sUbq system); G.R. and S.F.E. analyzed the results; G.R. and S.F.E. wrote the paper.

## Competing interests

The authors declare no competing interests.
