## [Peer Review File · Communications Biology]

Reviewers' comments:

Reviewer #1 (Remarks to the Author):

This manuscript presents an innovative and forward looking study of the interactome for TuMV proteins in Arabidopsis. Using a Y2H screen the authors were able to map all interacting partners for each viral protein with the exception of P1 and CP. For advice I understand that these can be difficult to express individually. For P1 its possible to obtain expression if the full C-terminal cleavage site is included. P1 cleaves its C-terminus and the model is that the C-terminus remains lodged in the active site following cleavage and this gives proper folding to the mature protein. That said instability has been seen for different potyviruses and so I am not concerned by the lack of data regarding P1 and CP. NIB has the highest number of interactions and 6K2 has the least with 11. Unfortunately the excel data was not provided with the article. I would love to see that data, but it does not prevent me from reviewing the script and I leave it to the editor to ensure its inclusion.

Its fascinating that the absolute level of differential gene expression seems to have an inverse relationship with the connectivity degree. The interactions for TuMV proteins are enriched in hub targets which are already well expressed. This is important as parrallel studies using transcriptomics may try to derive conclusions about interactions based on gene expression. Publishing this work becomes important to contextualize transcriptomic work that is being carried out using plant potyviruses. TuMV proteins appear to engage with proteins from different parts of the interactome in a way that appears to deliberately coordinate the perturbations of host factors.

I have a couple of questions. Give the high interactions involving NIB and VpG, are these intrinsically disordered proteins? Proteins with high contacts likely have IDRs and interact with other IDR proteins. This could lead to cytoplasmic or nuclear aggregation. It might be worth mentioning IDR in the explanation of interactions

Next, Figure 2 appears to be lacking heterologous controls for BIFC. I think it would be helpful to include the controls here.

Please elaborate the Figure legend for S1.

Reviewer #2 (Remarks to the Author):

A comprehensive physical interaction map between turnip mosaic virus and Arabidopsis thaliana proteomes

The authors generate a large-scale map between proteins of turnip mosaic virus and proteins of a host plant Arabidopsis thaliana. Starting from a normalized cDNA library from different tissues using mostly Y2H, but then also screening the viral proteins against an Arabidopsis cDNA library using the split-Ubq system to identify potential interaction partners that are membrane bound. The resulting dataset of 381 interactions is then supplemented with 18 literature interactions for the same viral protein. Using this dataset, the authors conduct a number of analyses including functional enrichment analysis, analyzing the topology of the network, integrating the network with transcriptomic information and selected biological validation in infection assays using genetic null host plants.

The paper is interesting, generally conducted in a thorough manner, and does address an important research gap. At the same time, there are some very substantial shortcomings, some of which are conceptual and may be due to lack of familiarity with important concepts in network biology, others may indicate a lack of rigor and tendency towards over interpretation of the evidence.

1. Title: There is a large body of literature that explicitly or implicitly deals with 'incompleteness' of biological network maps (see Yu et al., Science 2008; Venkatesan et al., Nat Methods 2010 and Arabidopsis Interactome Mapping Consortium, Science 2011). Even though the authors cite several of these papers, they still claim to have a 'comprehensive' interaction map. For such a claim it must be demonstrated that their combined assays have a 100% sensitivity, that their

screens are fully saturated and that positively all Arabidopsis proteins have been interrogated to saturation in an assays that enables detection of all their interactions. A more achievable alternative would be to revise their grandiose claims to better reflect the incompleteness of all interaction network maps. Please remove all descriptions of the dataset as being 'complete', 'comprehensive' or otherwise without gaps.

2. pg 2, ln 18: TGA may have a proviral function for TuMV. However, the study by Wessling et al, CHM, 2014) demonstrated that a host protein may have a pro-pathogenic function for one pathogen and an anti-pathogen function for another. Therefore it should be more precisely stated for which proteins the proviral function has been demonstrated and this not be inflated to a general claim.

3. pg 2, ln 19: a bunch of protein interactions do not translate into a 'comprehensive mechanistic description' – see 1, plus 'mechanistic description requires MUCH more evidence and data. Again, over grandiose claims.

4. pg 6, ln 5: please provide ref for 'significant crop losses'

5. pg 6/7: please describe how artifacts from spontaneous and constitutive autoactivators were identified. This is critical!

6. pg 7, ln 12: if most proteins have not been studied, how can it be asserted that Nib has the most specific function?

7. pg7, ln 14: the fact that their screen was not fully complete should have led the authors to realize that their dataset CANNOT be complete.

8. pg 8, ln 5: for any large scale dataset it is critical to assess the data quality for the full dataset. Therefore I have a serious problem with the validation of 'selected' targets. As these are strongly biased towards interactors that are likely valid, the conclusion about the dataset only amounts to 'there are validatable targets in the dataset' – still 90% of the targets may be artifacts, e.g. from autoactivators. It will be important to do additional experiments with a similarly sized subset that of interactors that was randomly picked. In addition, negative controls for each interaction are missing that preclude drawing conclusions even from the presented experiments. As BiFC is notorious to yield signal even with unspecific proteins, the controls with each interactor tested against empty vector or related, not identical proteins (e.g. family members) is critical. In line with the PRS/RRS concept in the aforementioned papers it would be useful to include a sizeable number of positive and negative controls to demonstrate the limitations of the validation assay.

9. pg8, ln 18: it appears that the authors have first added interaction partners from the literature and then conducted the GO term analysis. Obviously this is circular. In fact, addition of literature interactions may lead to additional problems. Certainly for some analyses, including the functional enrichment in this section, the lit data must be removed and the analysis only be conducted on the systematically acquired dataset.

Same analysis: how was multiple hypothesis correction done?

10. pg 8, ln 21: I do not see how the authors can go from a GO enrichment analysis to conclusions that 'this [what?] illustrates the conflict between host [...] and virus'. Such conflicts may be there, but as a conclusion from a GO analysis a much more cautious wording is appropriate.

11. pg 9, ln 3: interactions do not equal perturbations. Even though some interactions will lead to perturbations, the authors lack evidence for such at this stage.

12. pg 10, ln 4: there is no rewiring – please be precise

13. pg 10, ln 7: it is completely unclear what is meant by 'emergent communication channels between virus proteins'. What is meant by communication? Please rephrase considering that proteins are structural and enzymatic molecules and virus proteins ultimately mediate replication.

14. pg 10, ln 22: how does the 'virus reach a hub'. Please be precise and avoid anthropocentric phrasing.

15. pg 11, ln 16: nature does not 'intend' anything – please rephrase

19. pg 11, ln 19: please rephrase "ability to spread information". Please find an example what kind of Information spreading is meant here? Hormone biosynthesis?

20. pg 12, ln 3: the observed enrichment for high expression is likely a consequence of the used cDNA library. This could be addressed by sequencing the library, making sure that interactor identification is not correlated with representation in the library, and also showing that the library itself is not enriched for high expression transcripts. These controls are critical to substantiate the statement.

21. pg 12, ln 12: P-value? Effect size? If only noise, please delete this statement

22. pg 12, ln 14: trade-off is a term from evolutionary theory. What trade-off do you think was identified here? Please explain in detail the evolutionary balance. If you meant to refer to an

'anticorrelation', please phrase it as such.

23. pg12, ln 20: approaching the trade-off front? Please quantify what you mean.

24, pg 13, ln 2/3: please quantify these statements

25, pg 13, ln 20, 23: evidence is only provided for host proteins. If specific interactions and specific viral proteins are to be implicated in anti- or pro-viral effects, the corresponding evidence must be provided.

26, pg 14, ln 2: evidence is only for TuMV. For such a broad or universal statement, please provide infection data with additional viruses (5 – 10). Alternatively rephrase according to the evidence.

27, pg 14: please explain briefly what HMP expresses.

28, pg 14, lns 8-10: it is unclear how accelerated replication and similar symptom severity is consistent with 'increased resistance'. Please elaborate.

29, pg 15, ln 7: please also compare against host proteins with a similar degree in a systematic network map – is the conservation related to protein degree of characteristic of viral host targets?

30., pg 15, ln 11: which host network was used to assess host degree? Please include host degree based on a systematic network map (AI-1) whenever possible, as the bias due to the hypothesis-driven character of the small scale literature can cause grave artifacts (Yu et al., Science 2008).

31. pg 15, ln 14-17; 19-21: please provide interpretation/conclusion from these observations.

32. pg 15, ln 22-24; pg 16, ln 3: given that viral infections constitute a selective pressure that should favor differentiating evolution, this observation is very counterintuitive and requires a brief discussion? Or is this a consequence of artifactual data due to autoactivators?

It may be helpful to read Wessling et al., CHM 2014.

Reviewer #3 (Remarks to the Author):

COMMSBIO-22-2189 "A comprehensive physical interaction map between turnip mosaic virus and *Arabidopsis thaliana* proteomes"

This manuscript provides information of a high throughput yeast two-hybrid system screening for *Arabidopsis* host factors interacting with turnip mosaic virus (TuMV) proteins. No interactions were found for TuMV P1 and coat protein. The putative host interaction partners identified were compared to those presented in literature. In addition to known interactions some novel ones could be suggested. The authors

While this study seeks to understand the overall disruptions that viral infection causes in the host cell, I think the conclusions are overgeneralized. I cannot see the direct links between interactions in Y2H and the following statements in the abstract:

- Host targets participate with a higher number of infection-related functions. Does this refer to the interactions of a single host protein with few viral proteins? Still, both interactions may serve one infection-related function.

- I don't agree with the statement about a comprehensive mechanistic description of a plant virus – host interplay, with potential impact on disease etiology. Some of the interactions are crucial for infection, while others have a minor effect. Other way around, some of the interactions are crucial for the host plant to survive/grow normally etc, while others have a minor effect. When the virus co-opts a host factor, the function of the host factor may change. Depending, if the host protein is a limiting factor or not, its host-specific function may or may not be disturbed. Finally, the same set of virus proteins may induce very different host responses, depending on how the virus is recognized by the host.

I drew attention to the fact that the interactions known from the literature presented in dataset 2 have all been validated by Y2H. If interactions have been found by other methods, are they missed by Y2H? The authors take the Argonaute 1 as an example and state that the interaction with HCPro may be indirect. However, it is likely that the interaction is direct with a specific binding site in HCPro (Pollari et al., 2020). The authors do not present the data about Y2H. The limits of the method could be discussed more critically. Although 25 of the interactions are validated by BiFC, some of the other interactions may turn out to be false positives. Interpretation of the data should be very critical.

To provide putative host factors for further characterization by Y2H is not original. However, I think this study provides information about what kind of host proteins have become selected as targets for viral proteins – stable proteins with many connections and important host functions, which makes sense from the point of view of the virus being successful.

Minor:

Page 4 lines 21-25: Please, clarify the use of the terms (genes, protein targets, and proteome) in this sentence.

Fig. 1 – While this figure illustrates the number of the interactions found, I don't find it very informative since it does not contain any information about the names of the host proteins.

Instead, datasets 1 and 2 are informative.

I don't understand the sentence on page 7 lines 9-11, which states that out of 381 interactions 378 were novel.

Page 8 and Fig. 2: please, indicate which of the 25 validated interactions were novel and how many of them were already validated in literature. This would make it easier for the reader.

Figure 3c) and the related text on page 9 lines 13-20: I don't understand what has been done here and what is shown in the image 3c. Please, carefully explain the experiment and the result and, please, provide more information in the figure legend.

Page 10, lines 7 and 8: What is a communication channel? Could it be a protein complex that contribute to the infection?

Figure 4 a) I propose that this image would be substituted with a table, which shows both the database accession number and the name of the protein to make the data quicker accessible for the reader.

Figure 5e): the interaction between TGA1 and OBE1 is not indicated.

The infection analysis in Fig. 7: I cannot find a description how the incidence and progression of symptom severity was determined. Please, describe in detail so that the reader can evaluate the objectivity of the result. In methods section, I understand that n=24 for each mutant line of Arabidopsis – is this correct? The significance of the differences in disease progression was calculated. Could it be marked to the figure, which of the disease progress curves in Arabidopsis mutants show statistically significantly altered course of infection.

Discussion:

Page 17, lines 15-18: Recent literature describe works where the route of the replication complex from ER towards chloroplasts via an unconventional Golgi pass by pathway and the host proteins involved has been investigated. Is any of the proteins found in the current study among them?

Page 17, line 21: P1 is expressed as part of the polyprotein – therefore, it is not correct to say that it is not expressed. CP is self-interacting, but so is also VPg and some other potyviral proteins.

Page 19: Much of the discussion is dedicated to R-gene mediated recognition of the virus and consequently activated PTI and ETI. NIB sumoylation enhances TuMV infection NIB transit from the nucleus to the cytoplasm and suppresses the NPR1 resistance pathway in the nucleus (Cheng et al., 2017). Please, discuss the NIB and NPR1 interaction considering what is known about the role of NIB in NPR1-mediated resistance.

Page 20: The hypothesis of linking P3-PIPO and aquaporin PIP 1;3 interaction to inhibition of H₂O₂ influx. Is there evidence of H₂O₂ influx in the used pathosystem? If this would be important, what could be the reasons that a mutation in this aquaporin does not affect the progress of infection?

In general, I would suggest going critically through the discussion and try to bind better the hypothesis of different interactions to what is already known of the molecular mechanisms used by the virus in TuMV – arabidopsis pathosystem.

Responses to Reviewer #1:

Q.1.1: Give the high interactions involving NIB and VpG, are these intrinsically disordered proteins? Proteins with high contacts likely have IDRs and interact with other IDR proteins. This could lead to cytoplasmic or nuclear aggregation. It might be worth mentioning IDR in the explanation of interactions

R.1.1: This is a great suggestion. We have looked for intrinsically disordered protein regions (IDPR) in all eleven TuMV proteins using the fIDPnn server (<http://biomine.cs.vcu.edu/servers/fIDPnn/>; Hu G, *et al.* 2021 *Nat. Commun.* **12**, 4438) and found evidence of disordered regions for VPg, but not for NIB (see new Fig. S2). It might be worth noting the two proteins for which we failed to identify interactors, P1 and CP, also show strong signals of IDPR (Fig. S2). Therefore, we did not observe a correlation between viral proteins that contain IDPRs and the number of interactors found in our screening. These results are now presented in the main text.

Q.1.2: Next, Figure 2 appears to be lacking heterologous controls for BIFC. I think it would be helpful to include the controls here.

R.1.2: These controls were obviously done but we omitted the pictures to keep the figure simpler since no fluorescence was seen in any of them. This is now indicated in the figure legend.

Q.1.3: Please elaborate the Figure legend for S1.

R.1.3: We have modified this figure legend, as requested.

Responses to Reviewer #2

Q.2.1: Title: There is a large body of literature that explicitly or implicitly deals with ‘incompleteness’ of biological network maps (see Yu et al., Science 2008; Venkatesan et al., Nat Methods 2010 and Arabidopsis Interactome Mapping Consortium, Science 2011). Even though the authors cite several of these papers, they still claim to have a ‘comprehensive’ interaction map. For such a claim it must be demonstrated that their combined assays have a 100% sensitivity, that their screens are fully saturated and that positively all Arabidopsis proteins have been interrogated to saturation in an assays that enables detection of all their interactions. A more achievable alternative would be to revise their grandiose claims to better reflect the incompleteness of all interaction network maps. Please remove all descriptions of the dataset as being ‘complete’, ‘comprehensive’ or otherwise without gaps.

R.2.1: We fully agree with this comment. We have modified the title. Also, we have avoided references to our PPI dataset as “complete” or “comprehensive” throughout the manuscript.

Q.2.2: pg 2, ln 18: TGA may have a proviral function for TuMV. However, the study by Wessling et al, CHM, 2014) demonstrated that a host protein may have a pro-pathogenic function for one pathogen and an anti-pathogen function for another. Therefore it should be more precisely stated for which proteins the proviral function has been demonstrated and this not be inflated to a general claim.

R.2.2: We don’t understand this particular concern. Indeed, we are not making any general claim in this sentence. We simply provide an example of a protein for which we have experimentally shown its proviral function in the case of TuMV. In any case, the sentence has been slightly rewritten to make it clearer.

Q.2.3: pg 2, ln 19: a bunch of protein interactions do not translate into a ‘comprehensive mechanistic discription’ – see 1, plus ‘mechanistic description requires MUCH more evidence and data. Again, over grandiose claims.

R.2.3: This sentence has been removed.

Q.2.4: pg 6, ln 5: please provide ref for ‘significant crop losses’

R.2.4: Done.

Q.2.5: pg 6/7: please describe how artifacts from spontaneous and constitutive autoactivators were identified. This is critical!

R.2.5: Library plasmids were rescued from clones identified as positives after the second round of screening and subsequently retransformed into yeast for validation of reporter gene activation, both in the presence and in the absence of the viral protein used in the screen. Only those clones failing to activate reporter gene expression on their own were selected for further analysis. This clarification has now been included in the text.

Q.2.6: pg 7, ln 12: if most proteins have not been studies, how can it be asserted that Nib has the most specific function?

R.2.6: From the virus perspective, NIB has not been described to have any function other than the RNA-dependent RNA polymerase activity and to interact with other viral and host factors to form viral replication complexes bound to ER membranes (for a recent review, see Shen *W et al.* 2020 *Viruses* **12**, 77). In this sense, NIB has a well-defined function: genome replication.

The sentence has been rewritten for clarity.

Q.2.7: pg7, ln 14: the fact that their screen was not fully complete should have led the authors to realize that their dataset CANNOT be complete.

R.2.7: The reviewer is absolutely right. Our dataset was incomplete as we were missing some interactions already described in the literature for TuMV. As stated above, this has been taken into account in the modifications of the manuscript.

Q.2.8: pg 8, ln 5: for any large scale dataset it is critical to assess the data quality for the full dataset. Therefore I have a serious problem with the validation of ‘selected’ targets. As these are strongly biased towards interactors that are likely valid, the conclusion about the dataset only amounts to ‘there are validatable targets in the dataset’ – still 90% of the targets may be artifacts, e.g. from autoactivators. It will be important to do additional experiments with a similarly sized subset that of interactors that was randomly picked. In addition, negative controls for each interaction are missing that preclude drawing conclusions even from the presented experiments. As BiFC is notorious to yield signal even with unspecific proteins, the controls with each interactor tested against empty vector or related, not identical proteins (e.g. family members) is critical. In line with the PRS/RRS concept in the aforementioned papers it would be useful to include a sizeable number of positive and negative controls to demonstrate the limitations of the validation assay.

R.2.8: We understand the concern of the reviewer. However, our ultimate aim with the BiFC was to validate a (small) random set of positive interactions with a different *in vivo* approach. This goal was achieved.

We fully appreciate the value of the PRS/RRS strategy, but unfortunately, we can’t perform additional experiments at this moment due to lack of specific funds. As we already explained in R.2.5, candidates that showed autoactivation of the reporter in the HT-Y2H screen were excluded. We have removed the last sentence of the paragraph to avoid the false impression our results are exhaustive and just present the results at their face value.

Q.2.9: pg8, ln 18: it appears that the authors have first added interaction partners from the literature and then conducted the GO term analysis. Obviously this is circular. In fact, addition of literature interactions may lead to additional problems. Certainly for some analyses, including the functional enrichment in this section, the lit data must be removed and the analysis only be conducted on the systematically acquired dataset. Same analysis: how was multiple hypothesis correction done?

R.2.9: We disagree that adding the interactions curated from the literature into GO analyses incurs in a circularity problem. We agree that literature curation could be error prone and of low quality, as previously pointed out by others (Cusik ME *et al.* 2009 *Nat Meth* **6**, 39-46). For sure it might not be exhaustive and biased towards interactions that result in some obvious phenotypic effect. However, from the perspective of completeness, we are convinced that ignoring virus partners that have been well established by others and missed in our screenings would provide a less precise picture of the real interactome. Notice that we have added only 18 interactors to a list of 378 experimentally determined ones, representing less than 5% of the final list used for the GO analyses. These 18 interactors were described for TuMV – *A. thaliana*. We have been conservative not including other interactions previously described for other potyviruses and plant hosts (e.g., *N. benthamiana*).

In any case, removing these 18 interactors does not qualitatively change the results presented in the manuscript. We have generated a new Fig. S3 that shows the results of the analyses without the 18 literature interactors. In terms of GO, response to stress”, “post-transcriptional regulation of gene expression”, “meristem development”, and “photosynthesis” remain as the most enriched functional categories.

Multiple hypothesis correction was done using Benjamini-Hochberg FDR, as already mentioned in the corresponding Methods subsection.

Q.2.10: pg 8, ln 21: I do not see how the authors can go from a GO enrichment analysis to conclusions that ‘this [what?] illustrates the conflict between host [...] and virus’. Such conflicts may be there, but as a conclusion from a GO analysis a much more cautious wording is appropriate.

R.2.10: The sentence has been reworded.

Q.2.11: pg 9, ln 3: interactions do not equal perturbations. Even though some interactions will lead to perturbations, the authors lack evidence for such at this stage.

R.2.11: We accept this suggestion. We have changed the word “perturb” by the more neutral “participate in”.

Q.2.12: pg 10, ln 4: there is no rewiring – please be precise

R.2.12: This sentence has been edited to express our idea more clearly.

Q.2.13: pg 10, ln 7: it is completely unclear what is meant by ‘emergent communication channels between virus proteins’. What is meant by communication? Please rephrase considering that proteins are structural and enzymatic molecules and virus proteins ultimately mediate replication.

R.2.13: We have removed this expression and explained what we meant. We have also exemplified our argument with two cases.

Q.2.14: pg 10, ln 22: how does the ‘virus reach a hub’. Please be precise and avoid anthropocentric phrasing.

R.2.14: The sentence has been rewritten.

Q.2.15: pg 11, ln 16: nature does not ‘intend’ anything – please rephrase

R.2.15: The “intentionality” has been removed from the sentence.

Q.2.16: pg 11, ln 19: please rephrase “ability to spread information”. Please find an example what kind of information spreading is meant here? Hormone biosynthesis?

R.2.16: We have rephrased this sentence. We provide one example of Ca²⁺-mediated hormonal orchestration in response to infection and oxidative stresses.

Q.2.17: pg 12, ln 3: the observed enrichment for high expression is likely a consequence of the used cDNA library. This could be addressed by sequencing the library, making sure that interactor identification is not correlated with representation in the library, and also showing that the library itself is not enriched for high expression transcripts. These controls are critical to substantiate the statement.

R.2.17: Again, sequencing the commercial library represents an effort that we can’t afford now. Furthermore, the manufacturer ensures that the library has been prepared reducing the copy number of highly expressed genes and increases the representation of low copy transcripts. Quoting from Takara’s website: “This yeast two-hybrid library was constructed from mRNA isolated from 11 *Arabidopsis* tissues, mixed in equal quantities and transformed into yeast strain Y187. The cDNA was normalized prior to library construction to reduce the copy number of abundant cDNAs derived from highly represented mRNAs, thereby increasing the representation of low copy number transcripts. The normalization process combines a Duplex-Specific Nuclease (DSN) treatment and SMART technology, reduces the number of clones that must be screened in your yeast two-hybrid assay, and facilitates the identification and characterization of novel protein-protein interactions. The library was transformed into yeast strain Y187 and can be readily mated to a MATa GAL4 reporter strain, such as AH109 or Y2HGold, for screening”. We have underlined the relevant sentence.

Therefore, we are confident that our analysis and conclusions are reasonably solid within the limitations associated to the technology employed. In order to clarify this important point, we have included this explanation in the corresponding Methods section.

Q.2.18: pg 12, ln 12: P-value? Effect size? If only noise, please delete this statement

R.2.18: The statement has been removed.

Q.2.19: pg 12, ln 14: trade-off is a term from evolutionary theory. What trade-off do you think was identified here? Please explain in detail the evolutionary balance. If you meant to refer to an ‘anticorrelation’, please phrase it as such.

R.2.19: In this case, we more precisely meant a negative correlation. We do not intend to make any evolutionary conclusion here.

Q.2.20: pg12, ln 20: approaching the trade-off front? Please quantify what you mean.

R.2.20: We have run a new analysis finding the Pareto front for the relationship between the expression of a gene and its degree in the APPIN network. The front corresponds to the set of values that simultaneously optimize the maximum connectivity degree and gene expression. Then, we computed the Euclidean distance between each viral partners and the Pareto front and chose to discuss those that show the smallest distance. Consequently, “trade-off front” has been changed to “Pareto front”.

Q.2.21: pg 13, ln 2/3: please quantify these statements

R.2.21: These statements are provided as a sort of take-home message of this section. The corresponding quantifications are presented within the section.

Q.2.22: pg 13, ln 20, 23: evidence is only provided for host proteins. If specific interactions and specific viral proteins are to be implicated in anti- or pro-viral effects, the corresponding evidence must be provided.

R.2.22: Unfortunately, we don't understand this comment. What we have shown here is the effect that mutating candidate host interactors has in the progression of infection. We are not sure what additional “corresponding evidence” the reviewer would like to see.

Q.2.23: pg 14, ln 2: evidence is only for TuMV. For such a broad or universal statement, please provide infection data with additional viruses (5 – 10). Alternatively rephrase according to the evidence.

R.2.23: OK, fair enough. We have modified the text to focus our conclusions on TuMV. Obviously, expanding these experiments to 5 – 10 more viruses at this time is absolutely beyond the scope of this manuscript.

Q.2.24: pg 14: please explain briefly what HMP expresses.

R.2.24: This statistical method was already explained in the corresponding Methods subsection and appropriate references provided. For clarity, we have spelled out *HMP* to harmonic mean *P*-value the first time the acronym is used.

Q.2.25: pg 14, lns 8-10: it is unclear how accelerated replication and similar symptom severity is consistent with ‘increased resistance’. Please elaborate.

R.2.25: We have rewritten these sentences. We hope our message is now clearer.

Q.2.26: pg 15, ln 7: please also compare against host proteins with a similar degree in a systematic network map – is the conservation related to protein degree of characteristic of viral host targets?

R.2.26: This is a really good suggestion. We have run a new analysis in which the distribution of p_N/p_S values observed for TuMV-targeted proteins is compared with 1000 random sets generated from non-targeted proteins with a connectivity degree five or more. Not surprisingly, the difference is not significant in this case, since we are now comparing two sets of highly connected proteins that are under strong selective constraints other than the those generated by the virus. This analysis is shown in the new Fig. S5.

Q.2.27: pg 15, ln 11: which host network was used to assess host degree? Please include host degree based on a systematic network map (AI-1) whenever possible, as the bias due to the hypothesis-driven character of the small scale literature can cause grave artifacts (Yu et al., Science 2008).

R.2.27: For all PPI network-based analyses, we used the union between AI-1_{MAIN} and PPIN-2 network models. For simplicity, we refer to this union as APPIN throughout the text.

Q.2.28: pg 15, ln 14-17; 19-21: please provide interpretation/conclusion from these observations.

R.2.28: Our aim here is just to mention an observation. We have no obvious interpretation for this observation.

Q.2.29: pg 15, ln 22-24; pg 16, ln 3: given that viral infections constitute a selective pressure that should favor differentiating evolution, this observation is very counterintuitive and requires a brief discussion? Or is this a consequence of artifactual data due to autoactivators? It may be helpful to read Wessling *et al.*, CHM 2014.

R.2.29: We assume the reviewer means “positive” or “diversifying” evolution. The observation is counterintuitive if one assumes that all virus-interacting proteins must behave as *e.g.*, members of the vertebrate MHC: virus drive the diversification of host defenses in a dynamic arms race. But a number of studies support the emerging view that many host proteins that interact with viruses (and pathogens, in general) are encoded by highly conserved genes.

Indeed, Wessling *et al.* was already cited in the previous version of our work. And yes, this fantastic work also shows that targets of ascomycetes, oomycetes and bacteria effectors are central proteins in the AI-1_{MAIN} network “which likely cannot tolerate much variation without adverse effects on protein function”. We now make this point explicit in our text.

Responses to Reviewer #3

Q.3.1: While this study seeks to understand the overall disruptions that viral infection causes in the host cell, I think the conclusions are overgeneralized. I cannot see the direct links between interactions in Y2H and the following statements in the abstract:

- Host targets participate with a higher number of infection-related functions. Does this refer to the interactions of a single host protein with few viral proteins? Still, both interactions may serve one infection-related function.

- I don't agree with the statement about a comprehensive mechanistic description of a plant virus – host interplay, with potential impact on disease etiology. Some of the interactions are crucial for infection, while others have a minor effect. Other way around, some of the interactions are crucial for the host plant to survive/grow normally etc, while others have a minor effect. When the virus co-opts a host factor, the function of the host factor may change. Depending, if the host protein is a limiting factor or not, its host-specific function may or may not be disturbed. Finally, the same set of virus proteins may induce very different host responses, depending on how the virus is recognized by the host.

R.3.1: Regarding the first statement. We have rewritten the sentence to clarify what we meant: targeted proteins were enriched in GO functions related to plant responses to infection.

The second controversial statement has been removed.

Q.3.2: I drew attention to the fact that the interactions known from the literature presented in dataset 2 have all been validated by Y2H. If interactions have been found by other methods, are they missed by Y2H? The authors take the Argonaute 1 as an example and state that the interaction with HCPro may be indirect. However, it is likely that the interaction is direct with a specific binding site in HCPro (Pollari *et al.*, 2020). The authors do not present the data about Y2H. The limits of the method could be discussed more critically. Although 25 of the interactions are validated by BiFC, some of the other interactions may turn out to be false positives. Interpretation of the data should be very critical.

R.3.2: Sorry we missed the Pollari *et al.* (2020) study. Having it in mind, perhaps HC-Pro and AGO1 was not the best choice to illustrate the point that affinity purification generates information about complexes, but not necessarily about binary contacts. We have selected another example from K. Mäkinen's group: HSP70 and its cochaperone CPIP interaction with CP.

This criticism is in line with most of the concerns raised by Reviewer 2 about false positives. Given the uncertainties of the study, we have toned down our conclusions and have dedicated a long paragraph in the Discussion to recognize these limitations.

Q.3.3: To provide putative host factors for further characterization by Y2H is not original. However, I think this study provides information about what kind of host proteins have become selected as targets for viral proteins – stable proteins with many connections and important host functions, which makes sense from the point of view of the virus being successful.

R.3.3: Thanks for your appreciation of the value of the information provided in this study.

Q.3.4: Page 4 lines 21-25: Please, clarify the use of the terms (genes, protein targets, and proteome) in this sentence.

R.3.4: The sentence has been rewritten.

Q.3.5: Fig. 1 – While this figure illustrates the number of the interactions found, I don't find it very informative since it does not contain any information about the names of the host proteins. Instead, datasets 1 and 2 are informative.

R.3.5: Yes. The figure only has illustrative purposes and we'd prefer to retain it. The full information can be found in Datasets S1 and S2. This is now indicated in the legend of the figure.

Q.3.6: I don't understand the sentence on page 7 lines 9-11, which states that out of 381 interactions 378 were novel.

R.3.6: We meant these interactions were described for the first time.

Q.3.7: Page 8 and Fig. 2: please, indicate which of the 25 validated interactions were novel and how many of them were already validated in literature. This would make it easier for the reader.

R.3.7: The 25 interactions validated by BiFC were all novel except the well-known interaction between VPg and eIF(iso)4E. The text has been rewritten to make this point clearer.

Q.3.8: Figure 3c) and the related text on page 9 lines 13-20: I don't understand what has been done here and what is shown in the image 3c. Please, carefully explain the experiment and the result and, please, provide more information in the figure legend.

R.3.8: We have rewritten this paragraph and modified the legend of Fig. 3 to better explain panel 3c.

Q.3.9: Page 10, lines 7 and 8: What is a communication channel? Could it be a protein complex that contribute to the infection?

R.3.9: Yes, precisely. Following the advice of Reviewer 2, we are not using the concept of "communication channel". We now illustrate the idea with two examples from Fig. 4a.

Q.3.10: Figure 4 a) I propose that this image would be substituted with a table, which shows both the database accession number and the name of the protein to make the data quicker accessible for the reader.

R.3.10: We prefer to maintain Fig. 4a as it is now. The database accession numbers and gene names can be found in Datasets S1 and S2. We direct the readers to these Datasets in the figure legend.

Q.3.11: Figure 5e): the interaction between TGA1 and OBE1 is not indicated.

R.3.11: The interaction between TGA1 and OBE1 is not included in the extended Arabidopsis interactome we are using in this work (APPIN).

Q.3.12: The infection analysis in Fig. 7: I cannot find a description how the incidence and progression of symptom severity was determined. Please, describe in detail so that the reader can evaluate the objectivity of the result. In methods section, I understand that n=24 for each mutant line of Arabidopsis – is this correct? The significance of the differences in disease progression was calculated. Could it be marked to the figure, which of the disease progress curves in Arabidopsis mutants show statistically significantly altered course of infection.

R.3.12: We now provide additional details on how disease progression was evaluated. Infection status of plants was determined by the presence of visible symptoms and symptoms severity quantified in a semi-quantitative scale described elsewhere (e.g., Fig. 1 in Butković A *et al.* (2021) *Virus Evol.* 7, veab063).

Yes, we inoculated 24 plants of each mutant line analyzed with TuMV and 24 extra plants were mock-inoculated.

We now indicate the significance level next to each pair of curves in Fig. 7. Notice that Fig. 7 shows one of the biological replicates while the statistics reported in the text are done for all sets of replicates.

Q.3.13: Page 17, lines 15-18: Recent literature describe works where the route of the replication complex from ER towards chloroplasts via an unconventional Golgi pass by pathway and the host proteins involved has been investigated. Is any of the proteins found in the current study among them?

R.3.13: We assume you're referring to the work by J.F. Laliberté group showing that TuMV replication vesicles bypass the Golgi and the implication of SNARE and COPII proteins. In our screening, we found that 6K2 interacts with the SNARE SEC22 protein. This interaction was already shown by García Cabanilles D *et al.* (2018) *Plant Cell* 30, 2594. We now discuss this process in the Results section. No other protein described in the context of vesicle trafficking has been found in our screenings.

Q.3.14: Page 17, line 21: P1 is expressed as part of the polyprotein – therefore, it is not correct to say that it is not expressed. CP is self-interacting, but so is also VPg and some other potyviral proteins.

R.3.14: The detail about P1 is now mentioned in the text.

We believe that one should be cautious about concluding that CP self-assembly is the only explanation for not finding its interactors. It might be just a matter of more or less propensity to self-assembly.

Q.3.15: Page 19: Much of the discussion is dedicated to R-gene mediated recognition of the virus and consequently activated PTI and ETI. Nib sumoylation enhances TuMV infection Nib transit from the nucleus to the cytoplasm and suppresses the NPR1 resistance pathway in the nucleus (Cheng et al., 2017). Please, discuss the Nib and NPR1 interaction considering what is known about the role of Nib in NPR1-mediated resistance.

R.3.15: Thank you for pointing out this very interesting interplay between NPR1, SUMO3 and Nib. We have elaborated on it in the Discussion.

Q.3.16: Page 20: The hypothesis of linking P3-PIPO and aquaporin PIP 1;3 interaction to inhibition of H₂O₂ influx. Is there evidence of H₂O₂ influx in the used pathosystem? If this would be important, what could be the reasons that a mutation in this aquaporin does not affect the progress of infection?

R.3.16: Yes, there is evidence of H₂O₂ influx during *A. thaliana* infection by TuMV. A first report by Kim B *et al.* (2008) *Mol. Plant Microbe Interact.* **21**, 260 showed that the HR response to TuMV infection was associated with the production of H₂O₂. More recently, Otulak-Kozziel K *et al.* (2020) *Int. J. Mol. Sci.* 21, 8510 showed that *A. thaliana* respiratory burst oxidase homologues (RBOH) were involved in the response to TuMV infection: *rboh* mutants had reduced H₂O₂ accumulation and enhanced PR1 activation, resulting in more resistant plants.

As we conclude now in the text, despite our expectation, that the interaction between PIP1;3 and P3N-PIPO might not be biologically relevant in terms of virus infection.

Reviewers' comments:

Reviewer #1 (Remarks to the Author):

I believe the authors have answered all concerns very well and the manuscript is acceptable to move forward.

Reviewer #2 (Remarks to the Author):

Review

A comprehensive physical interaction map between turnip mosaic virus and Arabidopsis thaliana proteomes

The manuscript is greatly improved and most of the unclear phrases pointed out in the first review have been clarified and are adequately precise now.

In the course of rewriting a few new phrases and issues came up though that require rewriting and possibly some reanalyses.

Pg4 ln 9: "which are otherwise difficult to reach by random attacks" – random attack is an artificial node removal that mimics an external attack. "Paths" and "reaching" are not fitting concepts here. Please rephrase.

Pg 4 ln 13: "less significantly than through pure centrality directed attacks" – please rephrase – specifically: do you mean "less significantly" (and what is that supposed to mean) or "significantly less". In fact, the entire sentence is unclear to me. Please revisit. Also make sure to not equal 'number and diversity of interactions' with 'perturbations' unless these are experimentally demonstrated.

Pg 5, ln 10: exhaustive goes back to the previous criticism of 'complete/comprehensive'. "systematic" analysis could be used as this is missing and one of the strengths of this manuscript.

Pg 7 ln 11: I suggest to stay away from 'sticky' which is often used in a derogative manner to imply artifacts. I suspect the authors do not aim to evoke this impression.

Pg 7, pgph ln 13: it would be appropriate to discuss the incompleteness of the interactor set in established terms of "assay sensitivity" and "sampling sensitivity" (Venkatesan et al, Nat Meth, 2010; Braun et al.; Nat Meth, 2010). Briefly, no single assay is able to detect all interactions and most pipelines require multiple repeat screens to even approach saturation of this inherent assay ceiling.

Pg7, lns 20 – 25: It is unclear why these statements are relevant for the narrative

Pg 9, ln 2: The last sentence refers to validated interactions shown on fig 2, right? Perhaps a rephrase like the following is a tad more positive: "All other validated interactions involve N1b, indicating that also the many interactions of this protein validate at a high rate."

Pg 9, ln 9: GO enrichment. This also goes to the response letter of the authors. The issue is inclusion of the known interactors from the literature in the GO analysis, which creates a circularity, which the authors disagree with. Let me explain: previously identified interactors have likely received relevant GO annotations, for example 'response to virus'. By including such prior data, the authors are likely to import host proteins that contribute to relevant annotations. I do agree that excluding previously described host proteins will also exclude relevant (known) biology and they should be included for non statistical hypothesis development. Nonetheless, for any statistical analyses, it is important to stick to the systematically collected datasets in order to avoid circularity and biases. As the authors have already repeated the analyses without the 18 interactions, it is only required to limit the discussion of the analysis and results to those terms that are significant in this corrected analysis.

Pg 9, ln 15: it is unclear how 'developmental process' illustrates 'the conflict between virus and host'. It would appear this illustrates the conflict between defense and development/growth within the host. Perhaps better example terms can be found, or rephrase.

Pg 9, ln 19: again, do not equate 'interaction' with 'perturbation'. A qualifying 'appear to perturb' would make it clear that the perturbation is expected based on the interaction, but has not been shown (at least at this point of the narrative).

Pg 9, ln 25: "appears" to impact more host functions – there is no functional evidence for an actual

impact on these functions by Nib.

Pg 10, ln 11: please revisit the P3 sentence and especially rephrase 'less related' in a positive way – 'most unique set of host targets'?

Pg 11, ln 4: 'additional interactions' additional to what? This sentence is unclear.

Pg 11, ln 18: In the response letter the authors explain that APPIN was composed of PPIN2 and AI-1, however here only the AI-1 paper is cited. This makes sense as in PPIN-2 no new host-host interactions are reported. Most importantly, however, the authors must limit the network analyses to the AI-1MAIN subset. Both, the PPIN-1 and the AI-1RPT datasets were done using small subsets of proteins and slight different experimental pipelines (fragments, different search space, more repeats) that introduce heavy biases into the network map. To illustrate: for the AI-1RPT dataset, a small subset of the search space was screened 6 times. Thus, for these proteins more interactions could be identified than for the rest of the search space and their degrees are inflated. These analyses need be redone. Also, please refer to the host dataset as AI-1MAIN to eliminate the risk for any misunderstanding.

How many TuMV proteins were contained in the network and how many remained without network connections in this systematic dataset?

Pg 12, ln 22: the point that TuMV targets proteins with higher connectivity was made several times on this page. Perhaps condense.

Pg 13, ln 24: replace APPIN by AI-MAIN in the analysis and text.

Pg 14, ln 13: Why do you refer to the interactors as 'predicted'? These were measured and the dataset was validated? This qualification can be removed to phrase more strongly.

Generally, this following section is much more cleanly formulated – very nice.

Pg 16: what is the interpretation of the lack of signal after degree correction?

Pg 17: For the second analysis it is just as important as before to control against proteins of a similar degree. Is the observed signal specific for TuMV targets, or just a property of the targeted subset, but unlinked to viral targeting? Here, the interpretation of the previous analysis is similarly important. If there is no unambiguous interpretation, because it is not possible to differentiate between the influence of degree-centrality or conservation on viral targeting, it would be helpful to state this explicitly.

Pg 18: it is not necessary to put 'limited' in the headline. This is a very powerful resource and should be advertised as such – the limitations should be stated in the honest discussion, but it not needed in the headline.

Reviewer #3 (Remarks to the Author):

The revised manuscript COMMSBIO-22-2189A is a substantially improved version. It reads well and the authors have answered all my concerns.

Response to Reviewer #1 remarks:

R1.Q1. I believe the authors have answered all concerns very well and the manuscript is acceptable to move forward.

R1.A1. Thank you very much for your useful first review and for supporting the acceptance of the manuscript.

Response to Reviewer #2 remarks:

R2.Q1. Pg4 ln 9: “which are otherwise difficult to reach by random attacks” – random attack is an artificial node removal that mimics an external attack. “Paths” and “reaching” are not fitting concepts here. Please rephrase.

R2.A1. The sentence has been slightly rewritten.

R2.Q2. Pg 4 ln 13: “less significantly than through pure centrality directed attacks” – please rephrase – specifically: do you mean “less significantly” (and what is that supposed to mean) or “significantly less”. In fact, the entire sentence is unclear to me. Please revisit. Also make sure to not equal ‘number and diversity of interactions’ with ‘perturbations’ unless these are experimentally demonstrated.

R2.A2. The sentence has been rewritten.

R2.Q3. Pg 5, ln 10: exhaustive goes back to the previous criticism of ‘complete/comprehensive’. “systematic” analysis could be used as this is missing and one of the strengths of this manuscript.

R2.A3. Thank you, we have replaced “exhaustive” by “systematic” as suggested.

R2.Q4. Pg 7 ln 11: I suggest to stay away from ‘sticky’ which is often used in a derogative manner to imply artifacts. I suspect the authors do not aim to evoke this impression.

R2.A4. Right! We did not want to evoke the impression of artifactual results. We have replaced “sticky”.

R2.Q5. Pg 7, pgph ln 13: it would be appropriate to discuss the incompleteness of the interactor set in established terms of “assay sensitivity” and “sampling sensitivity” (Venkatesan et al, Nat Meth, 2010; Braun et al.; Nat Meth, 2010). Briefly, no single assay is able to detect all interactions and most pipelines require multiple repeat screens to even approach saturation of this inherent assay ceiling.

R2.A5. Using the meager list of literature curated potyvirus – plant interactions as positive reference set, we have estimated that our assay sensitivity is ~69%. That is, out of the 58 interactions described by other authors using quite different methodologies, our assays successfully detected 40 of them.

Evaluating sampling sensitivity would require of repeating a number of screens to achieve saturation. Given we have not repeated the screening, and this is not a possibility right now, it is not possible to evaluate this parameter.

We have added some additional information about the library in the Methods section to show that each mRNA is represented, on average, by 400 clones.

Dear referee, please notice that Venkatesan et al. and Braun et al. both work with extremely rich and well annotated human PPI data. For this, type of studies and datasets, a PRS/RRS strategy is doable. However, the richness of information for plant - potyvirus interactions is orders of magnitude smaller, thus limiting our capacity to do certain type of experiments and tests. Indeed, the main value of our work is precisely to provide a FIRST systematic exploration.

R2.Q6. Pg7, lns 20 – 25: It is unclear why these statements are relevant for the narrative

R2.A6. We agree this text is somehow distracting. We have removed it.

R2.Q7. Pg 9, ln 2: The last sentence refers to validated interactions shown on fig 2, right? Perhaps a rephrase like the following is a tad more positive: “All other validated interactions involve N1b, indicating that also the many interactions of this protein validate at a high rate.”

R2.A7. Yes. Thank you for the suggestion; we have incorporated it.

R2.Q8. Pg 9, ln 9: GO enrichment. This also goes to the response letter of the authors. The issue is inclusion of the known interactors from the literature in the GO analysis, which creates a circularity, which the authors disagree with. Let me explain: previously identified interactors have likely received relevant GO annotations, for example ‘response to virus’. By including such prior

data, the authors are likely to import host proteins that contribute to relevant annotations. I do agree that excluding previously described host proteins will also exclude relevant (known) biology and they should be included for non statistical hypothesis development. Nonetheless, for any statistical analyses, it is important to stick to the systematically collected datasets in order to avoid circularity and biases. As the authors have already repeated the analyses without the 18 interactions, it is only required to limit the discussion of the analysis and results to those terms that are significant in this corrected analysis.

R2.A8. Very respectfully, we disagree with the argument of circularity expressed by the reviewer. From our understanding, circularity would be generated if we pulled out from the host proteome all proteins annotated with the “response to virus” tag, add them to our list of experimentally + literature interactors and then interrogate the back databases for functional categories. Obviously, this is not what we have done. Again, we added a very limited number (18) of TuMV - arabidopsis experimentally confirmed interactions to our list of 378 interactors. As already explained in our previous response, adding or not these proteins had no effect in the results of the functional enrichment analyses.

Let's put in a different way. Stretching the reviewer's argument, we should not include in the analysis ANY protein detected in our experiments, not only those curated from the literature, because by the fact that identifying them in a screening against viral protein baits, we already know they must be related to “response to virus”.

In any case, we have added a footnote stating that the results are robust to removing the 18 literature-curated interactors. We hope this will close this discussion.

R2.Q9. Pg 9, ln 15: it is unclear how ‘developmental process’ illustrates ‘the conflict between virus and host’. It would appear this illustrates the conflict between defense and development/growth within the host. Perhaps better example terms can be found, or rephrase.

R2.A9. Plants respond to infection by modifying their development. It is well known that TuMV infection induces a development arrest. Likewise, one of the most obvious symptoms of infection is chlorosis, which result from the malfunctioning of the chloroplasts. We have added extra text to contextualize this sentence.

R2.Q10. Pg 9, ln 19: again, do not equate ‘interaction’ with ‘perturbation’. A qualifying ‘appear to perturb’ would make it clear that the perturbation is expected based on the interaction, but has not been shown (at least at this point of the narrative).

R2.A10. Thank you for the clarification. We have changed the text as suggested.

R2.Q11. Pg 9, ln 25: “appears” to impact more host functions – there is no functional evidence for an actual impact on these functions by Nib.

R2.A11. Thank you for the clarification. We have changed the text as suggested.

R2.Q12. Pg 10, ln 11: please revisit the P3 sentence and especially rephrase ‘less related’ in a positive way – ‘most unique set of host targets’?

R2.A12. Thank you for the suggestion. We have modified the text accordingly.

R2.Q13. Pg 11, ln 4: ‘additional interactions’ additional to what? This sentence is unclear.

R2.A13. The word “additional” has been removed. “Interactions” shall be enough here.

R2.Q14. Pg 11, ln 18: In the response letter the authors explain that APPIN was composed of PPIN2 and AI-1, however here only the AI-1 paper is cited. This makes sense as in PPIN-2 no new host-host interactions are reported. Most importantly, however, the authors must limit the network analyses to the AI-1MAIN subset. Both, the PPIN-1 and the AI-1RPT datasets were done using small subsets of proteins and slight different experimental pipelines (fragments, different search space, more repeats) that introduce heavy biases into the network map. To illustrate: for the AI-1RPT dataset, a small subset of the search space was screened 6 times. Thus, for these proteins more interactions could be identified than for the rest of the search space and their degrees are inflated. These analyses need be redone. Also, please refer to the host dataset as AI-

1MAIN to eliminate the risk for any misunderstanding. How many TuMV proteins were contained in the network and how many remained without network connections in this systematic dataset?

R2.A14. Thank you for clarifying this issue to us. We refer now to AI-1_{MAIN} across the manuscript. All network analyses have been redone accordingly. Notice that, as you pointed out, PPIN-2 does not add additional host-host interactions to AI-1_{MAIN} and, consequently, results have not changed.

R2.Q15. Pg 12, ln 22: the point that TuMV targets proteins with higher connectivity was made several times on this page. Perhaps condense.

R2.A15. We agree with this comment. The text has been condensed.

R2.Q16. Pg 13, ln 24: replace APPIN by AI-MAIN in the analysis and text.

R2.A16. Done (see R2.A14 above).

R2.Q17. Pg 14, ln 13: Why do you refer to the interactors as ‘predicted’? These were measured and the dataset was validated? This qualification can be removed to phrase more strongly. Generally, this following section is much more cleanly formulated – very nice.

R2.A17. The word “predicted” has been removed.

R2.Q18. Pg 16: what is the interpretation of the lack of signal after degree correction?

R2.A18. This is a really good question. We have rewritten the text to make the point clear as well as provide an explanation.

R2.Q19. Pg 17: For the second analysis it is just as important as before to control against proteins of a similar degree. Is the observed signal specific for TuMV targets, or just a property of the targeted subset, but unlinked to viral targeting? Here, the interpretation of the previous analysis is similarly important. If there is no unambiguous interpretation, because it is not possible to differentiate between the influence of degree-centrality or conservation on viral targeting, it would be helpful to state this explicitly.

R2.A19. We did the same test than for the p_N/p_S analysis, *i.e.* sampling only proteins with degree ≥ 5 , and found the same effect. Therefore, the interpretation is non ambiguous and does not depend on the method to evaluate selection.

R2.Q20. Pg 18: it is not necessary to put ‘limited’ in the headline. This is a very powerful resource and should be advertised as such – the limitations should be stated in the honest discussion, but it not needed in the headline.

R2.A20. Thank you for your appreciation of the value of our dataset. We have modified the headline.

Response to Reviewer #3 remarks:

R3.Q1. The revised manuscript COMMSBIO-22-2189A is a substantially improved version. It reads well and the authors have answered all my concerns.

R3.A1. Thank you very much for your useful first review and for supporting the acceptance of the manuscript.